# Towards an Understanding of Large-Scale Biodiversity Patterns on Land and in the Sea

**DOI:** 10.3390/biology12030339

**Published:** 2023-02-21

**Authors:** Grégory Beaugrand

**Affiliations:** CNRS, Univ. Littoral Côte d’Opale, Univ. Lille, UMR 8187 LOG, F-62930 Wimereux, France; gregory.beaugrand@univ-lille.fr

**Keywords:** biodiversity, climate, theory, metal, determinism, randomness, biogeography, bioclimatology

## Abstract

**Simple Summary:**

Among such questions as the origin of the universe or the biological bases of consciousness, understanding the origin and arrangement of planetary biodiversity is one of the 25 most important scientific enigmas according to the American journal *Science* (2005). This review presents a recent theory called the ‘macroecological theory on the arrangement of life’ (METAL). METAL proposes that biodiversity is strongly influenced by the climate and the environment in a deterministic manner. This influence mainly occurs through the interactions between the environment and the ecological niche of species *sensu* Hutchinson (i.e., the range of species tolerance when several factors are taken simultaneously). The use of METAL in the context of global change biology has been presented elsewhere. In this review, I explain how the niche–environment interaction generates a mathematical constraint on the arrangement of biodiversity, a constraint called the great chessboard of life. The theory explains (i) why biodiversity is generally higher toward low-latitude regions, (ii) why biodiversity peaks at the equator in the terrestrial realm and why it peaks at mid-latitudes in the oceans, and finally (iii) why there are more terrestrial than marine species, despite the fact that life first appeared in the marine environment.

**Abstract:**

This review presents a recent theory named ‘macroecological theory on the arrangement of life’ (METAL). This theory is based on the concept of the ecological niche and shows that the niche-environment (including climate) interaction is fundamental to explain many phenomena observed in nature from the individual to the community level (e.g., phenology, biogeographical shifts, and community arrangement and reorganisation, gradual or abrupt). The application of the theory in climate change biology as well as individual and species ecology has been presented elsewhere. In this review, I show how METAL explains why there are more species at low than high latitudes, why the peak of biodiversity is located at mid-latitudes in the oceanic domain and at the equator in the terrestrial domain, and finally why there are more terrestrial than marine species, despite the fact that biodiversity has emerged in the oceans. I postulate that the arrangement of planetary biodiversity is mathematically constrained, a constraint we previously called ‘the great chessboard of life’, which determines the maximum number of species that may colonise a given region or domain. This theory also makes it possible to reconstruct past biodiversity and understand how biodiversity could be reorganised in the context of anthropogenic climate change.

## 1. Introduction

The discipline of biology covers all living systems, from the simplest organic molecules (molecular biology) to large biomes (biogeography), by crossing many organisational levels, such as cells, tissues, organs, species, biocoenoses, and ecosystems [1]. It is essentially a science of complexity (Box 1) [2,3]. Since the origin of life, whether on Earth or elsewhere [4], biological systems have constantly evolved to adapt to their environment [5]. Species have emerged or died at gradual or sometimes more sudden rates, apparent balances punctuated by periods when changes occur relatively quickly [6]. The variety of species is not only perceptible from a morphological or anatomical point of view but is also reflected in many life history traits (size, growth, lifespan) that influence reproduction and individual survival [7]. The diversity exhibited by the living is almost inexhaustible, and evolutionary tinkering may obscure a form of intelligibility that researchers aim to discover [8,9]. It is a subtle mix between chance and necessity [10]: chance because diversity finds its origin in the genetic variability maintained by mutations and intra and inter-chromosomal mixing, and necessity because there are fundamental limits, whether physical, genetic, physiological, or ecological.

Box 1Complexity in ecology and the scientific approach we have adopted to consider it within the framework of METAL.
**Complexity in biology**
+ Innumerable actors and factors.+ All elements are interconnected and interdependent.+ Multiple actions and feedbacks at different organizationallevels and spatio-temporal scales.+ Nonlinearity (threshold effect, hysteresis).+ Emergence of new properties that are difficult to predict fromproperties of the parts.
**How to deal with this complexity within the framework of METAL theory?**
+ The system is complex but it can be simplified at certain organizationallevels (consideration of emergent properties) and to somespatio-temporal scales (i.e. at the largest scales).+ At some organizational levels and spatio-temporal scales,the laws influencing the arrangement of biodiversity are simple.+ Non-linearity can be overcome (e.g. the concept of niche considerselegantly the non-linear responses of species to environmental fluctuations).+ The use of ecological properties at the organizational level and at therelevant spatio-temporal scales enables one to unify phenomena, patterns ofvariability and biological events that govern the arrangement of biodiversity.+ Their unification gives a high level of coherence to the phenomena andobserved events and improves their understanding and predictability.

The origin and evolution of biodiversity are now better known. Charles Darwin, and neo-Darwinism, laid the solid theoretical foundations [11,12,13,14,15]. However, there remains a fundamental question to be resolved. How is biodiversity and the species that compose it organised on our planet and how is the abundance or number of species modified in space and time [16]? These questions are fundamental because biodiversity strongly influences the functioning of ecosystems and thus regulates services such as atmospheric carbon dioxide sequestration, but also provisioning services, i.e., the exploitation of ecosystems [17,18,19,20]. Moreover, to understand how anthropogenic climate change will affect individuals, species, and biocoenoses, the essential prerequisite is (i) to understand how these biological systems are naturally organised and (ii) to identify cardinal factors and mechanisms responsible for the alterations to then anticipate the modifications caused by environmental changes.

In this review, I present the macroecological theory on the arrangement of life (METAL), a theory that proposes that biodiversity is strongly influenced by the climatic and environmental regime in a deterministic manner (https://biodiversite.macroecologie.climat.cnrs.fr; accessed on 1 February 2023). This influence mainly occurs through the interactions between the ecological niche of species *sensu* Hutchinson (i.e., the range of a species tolerance when several factors are taken simultaneously) and the climate and environment [17]. The niche–environment interaction is therefore a fundamental interaction in ecology that enables one to predict and unify (i) at a species level, local changes in abundance, species phenology, and biogeographic range shifts, and (ii) at a community level, the arrangement of biodiversity in space and time as well as long-term community/ecosystem shifts, including regime shifts [21,22,23,24,25,26,27,28,29,30]. This theory offers a way to make testable ecological and biogeographical predictions to understand how life is organised and how it responds to global environmental changes [26]. More specifically, I show how METAL helps in understanding (i) why there are more species at low latitudes than at the poles, (ii) why the peak of biodiversity is located at mid-latitudes in the oceanic domain and at the equator in the terrestrial domain, and (iii) finally, why there are more terrestrial than marine species, despite the fact that biodiversity has emerged in the oceans. METAL has not been tested on prokaryotes (Bacteria and Archaea) yet because the species concept is fuzzy in this group, being replaced by the concept of operational taxonomic unit (i.e., taxa defined by molecular data analysis) [31,32]. Moreover, the ecological niche of prokaryotes can be more diverse and extreme, especially for Archaea [33,34], and their geographical ranges can be wide [35]. Therefore, all ecological principles examined in this review are only relevant for eukaryotes.

## 2. Patterns of Variability in Nature

For millennia, humans have detected recurring patterns of variability in nature or cycles [17,36,37,38,39,40,41]. The multitude of environments that our planet conceals forces clades to adapt to the local conditions, a process that rapidly fills the niche space [42]. Biogeographic studies have provided compelling evidence that some species are present only in tropical environments, while others are exclusively found in temperate or polar regions [41,43,44,45,46]. For example, Figure 1 shows that the spatial distribution of marine zooplankton (here copepod crustaceans) exhibits distinct patterns of variability [47], some species are present in the cold Labrador current (*Calanus glacialis*), others essentially along the European continental shelf (*Candacia armata*) or in the waters of the north (*Paraeuchaeta norvegica*) or the south (*Clausocalanus* spp.) of the North Atlantic, or finally at the transition between these waters along the North Atlantic Current (*Metridia lucens*).

Bioclimatologists and ecologists have noted the existence of cycles where periods of high abundance alternate with periods of low abundance or even absence [37,40,48,49,50]. In temperate ecosystems (e.g., the North Sea), some species flourish in the spring; we speak of spring phenology. Others bloom in the summer; we then speak of summer phenology [49,50]. The presence of these recurrent patterns of variability in space or time suggests the existence of control mechanisms, whether autogenic or allogenic [50].

## 3. The Difficult Identification of Patterns in Ecology

It is more difficult than it seems to identify these patterns of variability in nature, and sometimes—especially on a small scale—it may seem that there are no rules governing the arrangement of biodiversity [17]. Take the example of the simulated distribution of individuals from a fictitious species in a hypothetical region (Figure 2).

If we identify the number of individuals in an imaginary geographical square of 100 × 100 m, the distribution of individuals in this square appears random because no pattern of variability is identifiable (Figure 2a; the blue squares 1 × 1 m represent an individual). If we now cover an area of 19 × 19 km (there are 19,000/100 squares of 100 × 100 m in the figure, i.e., 190 × 190 = 36,100 squares), the distribution of the total number of individuals in each square of 100 × 100 m remains unintelligible (Figure 2b). Now imagine that we can examine the distribution of the number of individuals of this same species in a large region of 1000 × 1000 km (there are in Figure 2c 1,000,000/100 squares of 100 × 100 m, i.e., 10,000 × 10,000 = 100 million): we now see a pattern of variability emerging. The species is more abundant towards the centre of the region (Figure 2c). Looking at the pattern by taking height, that is to say, from a local to a large spatial scale, allows us to precisely identify the contours of the spatial distribution of this fictitious species (Box 1). An ecologist, who often studies biological systems on a small scale, may conclude that there are no detectable patterns of variability and that, therefore, the distribution of individuals is random and does not obey any rules (Box 1). On the other hand, a biogeographer may conclude that there is a structure, which implies the existence of underlying control mechanisms. The problem arises if researchers from these different disciplines extrapolate their results from the small to the large scale or inversely. In such a case, an ecologist may conclude that there are no principles governing the spatial distribution of a species and a biogeographer may establish certain predictions that are likely to be challenged on smaller scales. We touch here on the burning problem of scaling at the origin of so much controversy [51,52,53]. Referring to the analogy of an ecological theatre made by Hutchinson [54], Wiens [53] said, “to understand the drama, we must view it on the appropriate scale”. Note that this phenomenon is also observed along the time dimension. It is therefore essential in the construction of all theories to specify its limits according to the spatio-temporal scales one considers [55].

## 4. Towards a Better Understanding of Principles of Biodiversity Organisation and Climate Change Biology

METAL (macroecological theory on the arrangement of life) has recently been proposed to connect a large number of phenomena observed in biogeography (spatial distribution of species, communities and biodiversity), ecology (phenology, gradual or abrupt changes in communities or biodiversity), paleoecology (past distribution of species, communities and biodiversity) and bioclimatology (biogeographic and phenological shifts, temporal changes in abundance and biodiversity at local or regional scales) [17,21,22,24,29,30,49,50,56] (https://biodiversite.macroecologie.climat.cnrs.fr; accessed on 1 February 2023).

The unification of these phenomena is obtained by using the concept of the ecological niche of Hutchinson [57,58], which constitutes the elementary macroscopic brick of the theory, giving meaning and coherence to all phenomena, patterns of variability or events cited above (Figure 3). METAL considers the fundamental niche (i.e., the niche without the influence of species interaction), and current models do not explicitly include the influence of biotic interaction yet [25,29]. The niche can be divided into five components: (i) climatic, (ii) physico-chemical, (iii) substrate, or trophic with (iv) dietary and (v) resource concentration components [23,25,49]. It integrates all environmental conditions where a species’ individual can ensure its homeostasis, grow, and reproduce. A species’ niche therefore includes phenotypic plasticity, encompassing polyphenism and reaction norm (i.e., a species niche integrates the niches of all individuals of that species).

Therefore, the niche–environment interaction is considered to be a fundamental interaction in biology that explains and unifies a large number of patterns observed in ecology, biogeography and climate change biology [26]. This occurs because the genome controls many processes at infraspecific organisational levels (e.g., molecular processes) that affect physiological and morphological traits that in turn influence individual performance and fitness and finally determine the ecological niche of a species [50] (Figure 3). The use of the niche makes it possible (i) to implicitly consider these infraspecific processes without having to model them and (ii) to integrate the emergence of new biological properties impossible to anticipate from the property of the individual parts when crossing one or several organisational levels (here from the molecular to the specific level)(Box 1) [59,60].

Also known as species distribution models (SDMs) or bioclimatic envelope models [61,62,63], METAL integrates ecological niche models (ENMs) in its framework. ENMs primarily focus on the realised niche, which is based on past or contemporary spatial distribution and some key environmental (including climatic) variables. They then use the realised niche to project the likely distribution of a species in the past, present or future. ENMs have been extensively applied to project future species spatial distributions in the context of global climate change [61,64,65,66,67,68,69]. METAL provides a robust scientific baseline for ENMs and shows that this niche approach can be extended to explain many different phenomena at different organisational levels and spatio-temporal scales [22,70].

The niche–environment interaction is crucial for explaining, unifying and predicting a very large number of phenomena, patterns of variability or biological events observed in nature (Figure 4a) [26]. At an individual organisational level, the niche–environment interaction controls a large number of physiological and behavioural responses, such as the phenomena of thermotaxis and chemotaxis (Figure 4b) [17]. At a population organisational level, the niche–environment interaction controls species’ phenology and long-term changes in local abundance, including the arrival or extirpation of individuals of a species in a given area (Figure 4c) [49,50]. At a specific organisational level, the niche–environment interaction controls the distributional range of a species and even its extinction (Figure 4d) [23].

At a community level, the niche–environment interaction helps in understanding how communities are formed and modified, thus providing a theoretical basis for synecology and phytosociology (Figure 4e) [49,50]. The theory explains the seasonal succession observed in the marine planktonic environment, the gradual or abrupt modifications in communities, the biogeographical changes of biocoenoses (or assemblages), their contractions or expansions, and their eventual disappearance (Figure 4e) [23,28]. We can thus explain and anticipate major biological changes but also understand how biodiversity is organised and how it can be altered in the context of climate change [27,56].

Note, however, that human activities now influence a large number of processes that can interfere with the niche–environment interaction. For example, the extinction of a species or its long-term changes can be explained by anthropogenic pressures such as fishing, land use, hunting and pollution, pressures that are not further mentioned here [25,71,72,73]. A METAL model has recently considered fishing pressure and the niche together to explain the long-term changes in cod spawning stock biomass in the North Sea since the beginning of the 1960s [25].

The use of METAL in the context of climate change biology has been presented elsewhere [26]. In this review, I show how the niche–environment interaction generates a mathematical constraint on the large-scale arrangement of biodiversity and explains why there are more species on land than in the marine realm. To make progress on these questions, the scientific community continues to collect and inventory species and to study their biology [74,75]. A study suggested that the number of terrestrial and marine species could be 8,740,000 and 2,210,000, respectively [76]. Because the scientific team estimated that 1,233,500 species had been inventoried in the terrestrial environment and 193,756 in the marine environment (bottom and surface), this suggests that between 9% (marine) and 14% (terrestrial) of species have been named and described so far. (Note, however, that there exist many estimations in the scientific literature [75,77,78,79].) Meanwhile, ecologists continue to investigate the multiple interactions of these species with the environment, including the climate, but also biotic interactions, an essential prerequisite for understanding their spatial distributions (biogeography), their temporal patterns of reproduction (phenology) and their fluctuations from seasonal to centenary and millennial, as well as changes occurring on a geological time scale (ecology, palaeoecology and bioclimatology) [56,80,81,82,83,84,85,86]. With such a poor fundamental knowledge on species’ biology, how can we understand how factors and processes affect large-scale biodiversity patterns and design models to reconstruct them?

## 5. Large-Scale Biodiversity Patterns

### 5.1. A Brief Overview of the Main Hypotheses or Theories That Have Attempted to Explain Large-Scale Biodiversity Patterns

Why do some regions of the globe have more species than others? Among such scientific questions as the origin of life, the biological basis of consciousness or the composition of the universe, this question was cited as one of the 25 most important enigmas by the American journal *Science* in 2005 [16,87]. Indeed, for most taxonomic groups, it has been noticed that warm regions contain a higher number of species than polar regions [41,88,89,90,91,92]. Biogeographers generally speak of latitudinal biodiversity gradients to describe the large-scale biodiversity patterns observed in nature. The plural is important because the gradient may be different from one taxonomic group to another and from one domain to another (e.g., terrestrial and marine) [41]. For example, a maximum is obtained at the equator for a large number of terrestrial taxonomic groups, while it is rather subtropical for most oceanic taxonomic groups [41,88]. Although the existence of this biogeographical pattern has been known since Alexander von Humboldt in 1807 while he was in Central America and Charles Darwin after the return of the second expedition of the HMS *Beagle* in 1836, and that many hypotheses have been formulated for decades, no consensus has been reached [93,94,95,96,97,98,99,100,101].

What causes the latitudinal gradient in biodiversity, whether on land or in the sea, has been a topic of debate for decades, and more than 20 hypotheses or theories have been proposed [41,101,102,103,104,105,106,107,108,109]. It is beyond the scope of the present paper to review and discuss all of them, and below I only briefly review the main hypotheses or theories. While some authors have propounded that the biodiversity gradients are related to the larger area of the tropical belts [96,110], others have proposed null models of biodiversity, such as the neutral theory of biodiversity and biogeography [99] and the mid-domain effect (MDE) [111,112]. Moreover, it has been suggested that time is an important factor because speciation needs it to operate [90,113,114,115,116,117]. The tropics may assemble more species over a longer time period because they are more climatically stable than higher latitudes [95], and studies have provided evidence that the tropics are both a species cradle (higher origination rates) and a museum (more long-term climatic stability) [118,119]; See Vasconcelos and colleagues [120], however. Some studies have suggested that richer taxa have quicker diversification rates [121] and the metabolic theory of ecology predicts that the molecular clock is affected by body mass and temperature through metabolism [122].

Another popular hypothesis invokes the positive role of energy on biodiversity [95,123,124]. The energy hypothesis is frequently divided into two [123]: (i) exosomatic energy, where climatic factors such as temperature, precipitation and photosynthetically active radiation positively affect biodiversity, and (ii) endosomatic energy, which is the level of energy contained in the biomass that affects individuals and therefore the number of species. The latter hypothesis may be tested by using chlorophyll concentration or primary production [17]. Climate stability has also been invoked to explain the higher biodiversity in the tropics [125], along with magnitude, severity and frequency of environmental perturbations that are thought to limit species richness in temperate and polar regions [126,127]. In space, environmental heterogeneity promotes higher biodiversity [128,129]. For example, island species richness is positively correlated with habitat diversity [129]. The niche-assembly theory posits that there is more species richness in the tropics because there are more ecological niches, the niche being defined in term of resources [130]. Some hypotheses have invoked biotic interaction as a cause of speciation and therefore high species richness [131]. For example, Emerson and Kolm have provided evidence that the proportion of endemic species in an island covaries positively with biodiversity, suggesting that species richness increases speciation [17,131,132,133]; see, however, [134,135]. The argument seems tautological to some authors in terms of the search for the primary cause of these large-scale biodiversity patterns [17,95].

Perhaps the most compelling hypotheses are those that invoke an environmental control of biodiversity, such as environmental stability or energy availability [88,136,137]. Climatic hypotheses have been frequently proposed because large-scale biodiversity patterns correlate well with environmental parameters [88,137]. Among hypotheses, it has been suggested that global climate change may have shaped the large-scale patterns of biodiversity prevailing on Earth today because most clades originated in warm habitats, as temperatures have been predominantly warm during its history [138]. This hypothesis is known as tropical niche conservatism (TNC) [139]. Temperature has often been suggested to explain large-scale patterns in the distribution of marine organisms [88,140,141]. However, the exact mechanisms (e.g., metabolic theory of ecology [98]) by which the parameter may influence large-scale biodiversity patterns remain uncertain [100,101,102]. Finally, many authors have also suggested that many causes or factors interact to shape large-scale biodiversity patterns [30,142,143].

### 5.2. Modelling Biodiversity in METAL

Understanding the spatio-temporal arrangement of biodiversity on a large scale requires the development of numerical models where biological, environmental and climatic knowledge are put into equations [27,29,30,100]. In the context of the application of METAL, the fundamental bases of the biodiversity model are simple [100]. A large number of fictitious species is generated. Each fictitious species (called hereafter a pseudospecies) has unique physiological preferences that define their ecological (fundamental) niches, that is to say, their responses to climatic and environmental variability [23]. We can initially consider a simple niche, considering only the bioclimatic dimensions temperature and water availability (here precipitation). Temperature is an essential factor controlling the physiology of all species living on our planet and precipitation is a proxy for water availability, a variable just as important as temperature for terrestrial species. These climatic dimensions are fundamental, and many studies have underlined their importance [46,88,140].

Figure 5 shows an example with two marine pseudospecies, one being more eurythermal (i.e., tolerating a greater range of thermal variation) and the other being more stenothermal (i.e., tolerating a smaller range of thermal variation). In this example, we see that the stenothermal pseudospecies is characterised by a more limited range and lower abundance than the more eurythermal pseudospecies [21,23].

We can thus create a multitude of pseudospecies by varying the optimum and the ecological amplitude (i.e., niche breadth) of each niche dimension. Figure 6 shows the creation of marine pseudospecies from a simple Gaussian thermal niche [21,23]. Note that different types of niches can be used: from rectangular to trapezoidal [25,100] and from logistic to beta distribution [25,27], symmetrical or asymmetrical [25], parametric or nonparametric [23]. Moreover, the niche can be multidimensional [144], including nutrients, solar radiation or mixed-layer depth for phytoplankton, bathymetry and sediment types for fish, soil pH and composition for plants [25,65,66,144,145,146]. So far, most METAL simulations have been based on niches that vary between 0 (i.e., absence of a species for a given environmental regime) and 1 (i.e., highest abundance, or presence in case of a rectangular niche). Therefore, all species can reach the same level of maximum abundance. Although this assumption may possibly hold for a clade composed of species with a similar size, this is not so for a group that exhibits large size variability (e.g., mammals) [147,148,149]. Note, however, that this assumption does not affect biodiversity when the selected indicator is species richness (see below).

Different thermal optima and amplitudes are used [21,23]. In this example, when distributional ranges originating from one thermal niche are spatially separated, it is considered that they represent different species; therefore, one niche can give several species, in agreement with Buffon’s law, also known as the first principle of biogeography [41]. We see that thermal niches with lower thermal amplitudes give more species, although they exhibit smaller distributional ranges (Figure 6, left maps vs. right maps). Figure 5 shows that there is a relationship between the average abundance of a species and its area of distribution, a relationship already demonstrated empirically by Brown [150]. We extended this relationship by indicating that there is a positive link between the ecological amplitude of a species, its average abundance and its distribution area [23] (Figure 5 and Figure 6). These relationships hold for species of the same size [147,148] and trophic guild.

Examples from Figure 6 show that a niche can lead to more pseudospecies in the Northern than in the Southern Hemisphere (Figure 6b–d). This is due to the current location of continents that act as a barrier against gene flux, triggering more allopatric speciation in the Northern than the Southern Hemisphere (towards high latitudes). When the thermal amplitude is larger, the pseudospecies are more eurygraph and a single niche leads to fewer pseudospecies, e.g., only one in Figure 6a in each hemisphere. Moreover, the current configuration (i.e., south to north configuration) of the continents also enables more pseudospecies to emerge in the tropics (Figure 6g–h), especially when the pseudospecies are stenoecious and therefore stenograph (Figure 6e,f). Note that parapatric and sympatric speciations are not accounted for in this example. Allopatric speciation is thought to be a widespread mode of speciation in the marine environment, despite more evidence that other modes of speciation might also play a role [151]. Parapatric speciation is thought to be possible in the ocean [152,153,154]. Clinal parapatric speciation has been suggested for salps and some benthic species [151,155]. Sympatric speciation might also be frequent for marine invertebrates [156].

To reproduce the large-scale arrangement of biodiversity, we can build a model that first creates millions of niches, which then allow pseudospecies to establish themselves in a given region as long as environmental fluctuations are suitable [17,29,30,100]. The principle of the model is simple. It starts to create a large number of niches where both optima and amplitudes with respect to temperature only for the marine realm and both temperature and precipitation for the terrestrial realm vary. Many niches (i.e., with all possible optima and amplitudes), which can also overlap, are created (i) for temperature between −1.8 °C and 44 °C in both realms and (ii) for precipitation between 0 and 3000 mm in the terrestrial realm only. The full procedure is described in [29]. At the end of the procedure, there are a maximum of 101,397 and 94,299,210 niches in the marine and terrestrial realms, respectively. About 25% and 1% of these niches are chosen randomly to perform the simulations in the marine and terrestrial realms, respectively [29]. The use of fictitious niches and species is especially useful, since we have only inventoried 9% of marine and 14% of terrestrial biodiversity and we know little about the biology of most species (see Section 5.1). A niche can give rise to several pseudospecies if individuals from different regions never come into contact (e.g., Figure 6f) [29]. Pseudospecies are gradually colonising the terrestrial and marine environment (surface and bottom). During the simulations, the species organize themselves into communities and the biodiversity. More precisely, here the number of species in a given region is reproduced.

Beaugrand and colleagues [29] used this approach to model the biodiversity of the terrestrial and marine realm, including the surface and the bottom of neritic and oceanic regions (Figure 7). These numerical experiments (or simulations) correctly reconstruct large-scale biodiversity patterns as they are observed nowadays for a large number of taxonomic groups in the terrestrial and marine environment (e.g., crustaceans, fish, cetaceans, plants, birds) [29]. The biodiversity maps for the ocean floors (Figure 7c,f) remain provisional, as few observations have been made to date to confirm these predictions [29]. The model also reproduces well past biodiversity patterns of the Last Glacial Maximum and the mid-Pliocene (e.g., foraminifera), as well as the Ordovician (e.g., acritarchs) [27,56].

## 6. The Great Chessboard of Life

The reconstruction of large-scale biodiversity patterns observed in nature is possible because the niche–climate interaction generates a mathematical constraint on the maximum number of species that can establish in a given region [30]. We have named this constraint the great chessboard of life (Figure 8) [30]. This particular chessboard has a number of geographical squares (i.e., wide squares in Figure 8) that correspond to different regions (marine or terrestrial). Note that these geographical squares are limited on the figure (i.e., 6 × 8 = 48 squares), but should be higher to correctly represent the variety of environments, e.g., one for every degree of latitude and longitude (i.e., 180 latitudes × 360 longitudes = 64,800 squares). Each square on the chessboard is composed of sub-squares (i.e., the narrow squares in Figure 8), which represent the number of climatic niches that determines the maximum number of species that can colonise a square (i.e., a region or a wide square). Only one species can establish in a sub-square (i.e., a climatic niche) of the chessboard according to the competitive exclusion principle of Gause [157], thereby the more sub-squares (L) in a given region, the higher the maximum number of species that an area may contain (Figure 8). S is the number of species that a square (i.e., an area) actually contains. Therefore, L represents a fundamental limit (what I call here a mathematical constraint) on species richness, even if the actual number can still vary according to other processes (see below). The different pieces on the chessboard (e.g., king, queen, pawn) symbolize the different biological properties of the species (e.g., their differences in terms of life history traits, such as reproduction). Q represents niche saturation, with Q = (S/L) × 100. A saturation of 100% means that all niches or potential species that a square may contain are occupied. Biological (degree of clade origination) and climatic (repeated Pleistocene glaciations) causes influence the percentage of saturation of the niches in each geographical square so that there remains a degree of valence on the number of species present on the great chessboard of life [30].

The number of maximum niches fixes an upper limit on the number of species that can colonise a given region by speciation or immigration [30]. Few species can colonise areas located towards the minimum (e.g., −1.8 °C) and maximum limits (e.g., 44 °C) of temperature and precipitation (e.g., 0 and 3000 mm for precipitation) [29]. The choice of these minimum and maximum values in the METAL models is therefore important because it affects the results [100]. In marine polar areas, corresponding nowadays to the lowest limit of temperature (close to −1.8 °C), the number of species that can establish are fundamentally limited by L, since two species having the same niche cannot coexist at the same time and in the same place [157]. On the chessboard (Figure 8), the number of sub-squares is two in the wide square between the polar front and North Pole.

At low latitudes, since the theoretical upper limits are not observed nowadays (i.e., the upper limit for temperature is frequently fixed to 44 °C; for a justification of the threshold, see [100]), terrestrial biodiversity is maximum at the equator and marine biodiversity in subtropical regions (Figure 7) [27,29,56,100]. (I will come back to this point in Section 7). The great chessboard of life therefore suggests that there is more species richness in regions where there are more ecological niches (*sensu* Hutchinson [58]), providing compelling evidence for the niche-assembly hypothesis, although the niche in this hypothesis has been usually defined in terms of resources [130] (see Section 5.1).

The biogeographical constraints (i.e., low and high number of niches in high and low latitudes, respectively) imposed by the chessboard on biodiversity may be quickly detectable because clade diversification (not implemented in this METAL model) takes place relatively rapidly on a geological time scale [158]. Moreover, in the marine environment, taxa such as plankton have high dispersal capabilities [44,159,160] and may rapidly conform to the chessboard. Niche saturation (i.e., the number of observed species on the theoretical number of available niches) may help measuring the degree of conformity of the different taxonomic groups on the chessboard [30]. Although it is commonly assumed that niche saturation increases towards the equator (i) because evolutionary rates are thought to increase from cold to warm regions [122,161] and (ii) because of the presence of strong climate-induced environmental perturbations in extra-tropical regions that limit species richness [116,162], niche saturation is frequently highest towards the poles [30]. These apparently counterintuitive results suggest that the few sub-squares (i.e., the climatic niches) available on each region (i.e., wide square) of the chessboard are frequently occupied in polar regions (Figure 8). This means that low polar biodiversity should not always be attributed to low diversification rates (origination *minus* extinction) [114,115], but rather to a smaller maximum number of species’ niches (i.e., the parameter L on the chessboard in Figure 8) at saturation that locally limits biodiversity. This low number of niches over polar regions, and inversely the high number of niches equatorwards, originating from the niche–environment interaction, represent a mathematical constraint on the arrangement of biodiversity. Although there remains a great degree of freedom on the type and number of species that can establish in a region (e.g., origination and diversification of a clade), this number cannot exceed a threshold set by the niche–climate interaction.

In the marine realm, large-scale patterns of niche saturation differ among taxonomic groups, which suggests the existence of a particular chessboard for each group that might originate from taxon-specific diversification history [163,164]. For example, pinnipeds, which exhibit an inversed latitudinal biodiversity gradient, originate from Arctoid carnivores 25–27 Ma in the cold regions of the North Pacific [165]. Place of origin and time of emergence may therefore blur large-scale biodiversity patterns imposed by the great chessboard of life. Moreover, life history traits of each group make the great chessboard of life specific to a given taxonomic group, which sometimes explains the lack of universality of large-scale biodiversity patterns (Figure 8) [30].

The rate of diversification remains an important parameter because it determines the degree of niche occupation in a given geographical cell. Indeed, when polar regions are excluded, niche saturation of most groups (e.g., plankton and fish) but mammals is higher over permanently stratified regions [147]. Moreover, many clades should exhibit latitudinal biodiversity gradients towards the equator because their probability of emergence should be higher in the tropics, where there are more available niches, and palaeontological data have provided compelling evidence of greater rates of origination for tropical clades [166]—the hypothesis of tropical niche conservatism [139].

Beaugrand and colleagues suggest that the total number of species on the chessboard diminishes with organismal complexity [30], which can be explained by basic ecological and evolutionary processes. Endosomatic energy decreases from primary producers to top predators as a consequence of the second law of thermodynamics, decreasing the number of individuals and thereby species richness and niche saturation from producers to higher trophic levels [17,41]. Positive relationships between numbers of individuals and species richness has often been proposed to explain large-scale biodiversity patterns, e.g., the productivity theory [167], the area hypothesis [168], and the unified neutral theory of biodiversity and biogeography [99]. In addition to diminishing the number of individuals [169], a larger body also increases generation time [147], which slows down evolution [122,169]. Therefore, the likelihood that a taxon exhibits a large-scale biodiversity pattern different from the one imposed by the great chessboard of life is greater when its mean niche saturation is lower. This is especially the case for marine mammals. Pinnipeds, which have a low degree of niche saturation (<1% [30]), exhibit a pattern that does not conform to the great chessboard of life [30]. Note that it is possible to do specific simulations to account for the biology of a specific clade or taxonomic group, such as euphausiids, fish, coral reefs, or mangroves [29,145].

## 7. Differences in Latitudinal Biodiversity Gradients between the Terrestrial and the Marine Domains

A biodiversity peak is observed at the equator in the terrestrial realm and around the subtropical regions in the marine realm (Figure 7a–d). This distinction is related to the differential influence of atmospheric pressure fields in the marine and terrestrial realms [5,17,170,171]. Indeed, the high-pressure centres (i.e., the large-pressure high linked to the descending branches of the Hadley and Ferrel cells) provide climatic stability and heat, which increase surface biodiversity in the marine environment (Figure 9) [17]. However, above the continents, the high-pressure centres strongly limit precipitation, and biodiversity is therefore very low due to the lack of water availability [5,17,170,171]. The maximum values of biodiversity are reached in regions where precipitation is regular (towards the equator) and decreases when moving away from the influence of the intertropical convergence zone (ITCZ) [5], i.e., when it occurs a few weeks a year (monsoon areas). Finally, the cold ocean floors do not show a typical biodiversity gradient but a very homogeneous biodiversity pattern, except in high-latitude regions, where biodiversity decreases slightly, and over seamounts, where it is higher (Figure 7e–f).

The chessboard of life reorganizes when climate changes, which makes it dynamic from small to large temporal scales (i.e., geological scales) [30]. Indeed, large-scale biodiversity patterns are not stable over time [172], and METAL suggests that they were sometimes very different from those currently observed [56]. For example, during a cold period at the end of the Ordovician (510 million years ago), a very significant contrast probably existed between tropical biodiversity and the biodiversity of high-latitude regions (Figure 10a). Conversely, during the warm period of Stage 4 of the Cambrian (510 million years), the latitudinal gradient of biodiversity was probably reversed (Figure 10b). The current latitudinal gradient of biodiversity, characterised by a more or less regular increase in biodiversity from the poles to the equator, has therefore probably not always been observed since the appearance of eukaryotes on our planet [56]. Mannion and colleagues [172] also proposed that the current latitudinal gradient of biodiversity has not been a permanent feature through the Phanerozoic. They suggested that a biodiversity peak occurred during cold icehouse climatic regimes, whereas temperate peaks (or flattened gradients) were observed during warmer greenhouse regimes. In the context of current climate change, the use of METAL also suggests that the contrast between regions of low and high biodiversity may diminish towards the end of the century because a rise in temperature over permanently stratified regions (e.g., tropics and subtropics) reduces surface biodiversity, whereas an augmentation in temperature over temperate and polar regions increases biodiversity [27].

## 8. Why Are There More Terrestrial Than Marine Species?

At a high taxonomic level, the number of phyla of metazoans is higher in the ocean than on land, and this number is greater for the benthic than the pelagic realm [173]. In sum, 32 phyla are found in the sea and 21 are exclusively marine, whereas 12 are found on land, with only one being endemic to this realm (Onychophora) [173], while 27 phyla inhabit the benthos (with 10 endemic) and only 11 the pelagos (with one endemic—Loricifera). These estimates strongly suggest that diversification began in the sea, and probably in or close to the benthic realm [174]. However, at a species level, there are more terrestrial than marine species. Robert May assessed that ~85% of all species are terrestrial [173,175], Michael Benton ~75% [176], and more recent estimates suggest they may be closer to ~80% [76]. About 77% of animal species live on land, the remaining being found in freshwater (11%) and in the marine environment (12%), and 93% of plants live on land, 5% being freshwater and 2% marine [177]. Why are there more terrestrial than marine species? This conundrum is all the more incomprehensible, since marine biodiversity appeared long before terrestrial species [176,178]. Since more time has passed since the emergence of life in the oceans, the higher number of terrestrial species over marine species seems to be a counterintuitive observation.

METAL reproduces the difference in biodiversity observed between the marine and terrestrial domains well [29]. Although results depended upon the choice of the total number of niches, modelled biodiversity scaled to catalogued (and estimated) species gave 1,111,186 (8,825,091) for the terrestrial domain and 316,069 (2,242,908) for the marine domain. These estimates are close to those given in Section 4 [76]. Moreover, an estimate of the deep-sea benthic biodiversity (894,881 benthic species in areas below 2000 m and 256,278 in areas between 2000 m and 200 m) is close to what has been calculated in some studies [179,180]. Species density is expected to be higher over the shelf (200–2000 m) than deep sea, but because the latter realm is larger (301 vs. 36 million km^2^), there are more species in the deep-sea benthic realm [29].

Two mechanisms may explain why METAL reconstructs the difference between land and sea biodiversity well [29]. Firstly, Beaugrand and colleagues [29] suggested that the difference between the number of climatic dimensions among terrestrial and marine environments is fundamental. Water is evidently present everywhere in the ocean, which is not the case in the terrestrial environment. The additional discriminating climatic dimension in the terrestrial environment (water availability) arithmetically increases the number of climatic niches and thus the number of species that can establish in the terrestrial environment [29]. The addition of one climatic dimension increases the number of potential niches by ~100 [29].

Secondly, but to a lesser extent, Beaugrand and colleagues [29] suggested that the addition of a supplementary climatic dimension, combined with more pronounced geographical variations in the terrestrial environment (i.e., an increase in habitat heterogeneity), further fragment the geographical range of a species [181], increasing the possibility of allopatric speciation, i.e., the creation of species by prolonged or permanent geographic isolation of populations [29,41,182].

To conclude on this part, there are more terrestrial than marine species because there are more available climatic niches on land. This additional dimension inflates considerably the maximum number of niches and thereby species on the great chessboard of life (i.e., parameter L in Figure 8). This second fundamental dimension inflates considerably the number of species that the terrestrial realm may contain. The addition of the water availability dimension in the terrestrial realm (in addition to temperature) also morcellates species spatial distribution and increases the effect of landscape heterogeneity and the possibility for allopatric speciation [29]. In the ocean, the seascape is more uniform because there is only a single climate dimension (temperature). See, however, Ref. [181] for other important environmental dimensions. This influence is more prominent in the pelagic than the benthic environment, so it probably explains why there are more benthic than pelagic species [29]. The influence of seascape heterogeneity strongly affects local biodiversity over seamounts and shelves [183]. Therefore, local biodiversity should be higher over these areas, including heterogeneous shallow ones [29].

## 9. Conclusions

A central objective of biology and its sub-disciplines (e.g., biogeography, ecology) is to reveal the laws or general principles that govern the arrangement of life, but the sources of variations and exceptions seem inexhaustible. However, simple laws have been discovered in other areas of science, such as physics. Galileo Galilei wrote “*Philosophy [nature] is written in that great book which ever is before our eyes—I mean the universe—but we cannot understand it if we do not first learn the language and grasp the symbols in which it is written. The book is written in mathematical language, and the symbols are triangles, circles and other geometrical figures, without whose help it is impossible to comprehend a single word of it; without which one wanders in vain through a dark labyrinth*”. We show here that the niche–environment interaction is fundamental because it controls a large number of phenomena, patterns of variability, and biological events. In this review, I only show how METAL can help in understanding the arrangement of biodiversity, but the theory also explains other phenomena, such as spatial range, biogeographical shifts, phenology, annual plankton succession, long-term changes in species abundance, and community composition, gradual or abrupt [26].

Like the Italian scholar of the Renaissance Galileo Galilei, I propose that the great book of life is also written in mathematical language. In particular, the niche–environment interaction, controlled in part by the climatic regime, generates a mathematical constraint on the large-scale arrangement of biodiversity. We have named this constraint the great chessboard of life (Figure 8). The mathematical effect is probably considerable such that an inverted latitudinal gradient is impossible under present climatic conditions for most taxonomic groups that presently exhibit increasing biodiversity from the poles to the equator. Moreover, a similar mathematical effect explains why there are more terrestrial than marine species, even if the number of phyla is higher in the marine than terrestrial realm. The establishment of a global theory of biodiversity, however, requires taking into account a large number of biological processes that also influence biodiversity (e.g., diversification rate and origination place of a clade), and Theodosius Dobzhansky was greatly inspired when he wrote his article ‘Nothing in biology makes sense except in the light of evolution’ [12]. In addition to other key ecological factors discussed in Section 5, METAL should therefore consider more explicitly some key evolutionary processes in the future. In the process of developing such a global theory of biodiversity, considering all the complexity of biological systems (Box 1), it is important to recognize that mathematical constraints caused by (i) the number of key dimensions that the niches include in the terrestrial and marine realms and (ii) the niche–environment interaction also control the arrangement of biodiversity.

## Figures and Tables

**Figure 1 biology-12-00339-f001:**
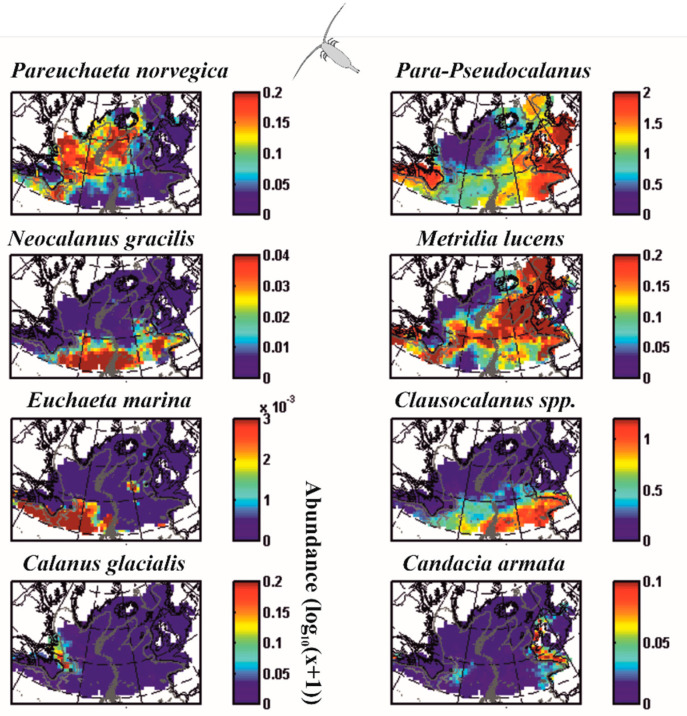
Mean spatial distribution of some marine copepods (planktonic marine crustaceans). Maximum abundance values are in red and zero abundances are in dark blue. The absence of colour corresponds to an absence of sampling. Some copepods are present in the icy or cold waters of the North Atlantic Ocean (*Pareuchaeta norvegica* or *Calanus glacialis*). Others occur in subtropical waters (*Clausocalanus* spp., *Neocalanus gracilis* and *Euchaeta marina*). The *Para-Pseudocalanus* group is present in temperate waters, *Metridia lucens* at the limit between temperate and cold waters, and *Candacia armata* mainly south of the European continental slope. These examples show that the distribution of species is not random on a large scale and that there are therefore control mechanisms. Redrawn, from Beaugrand and colleagues [47].

**Figure 2 biology-12-00339-f002:**
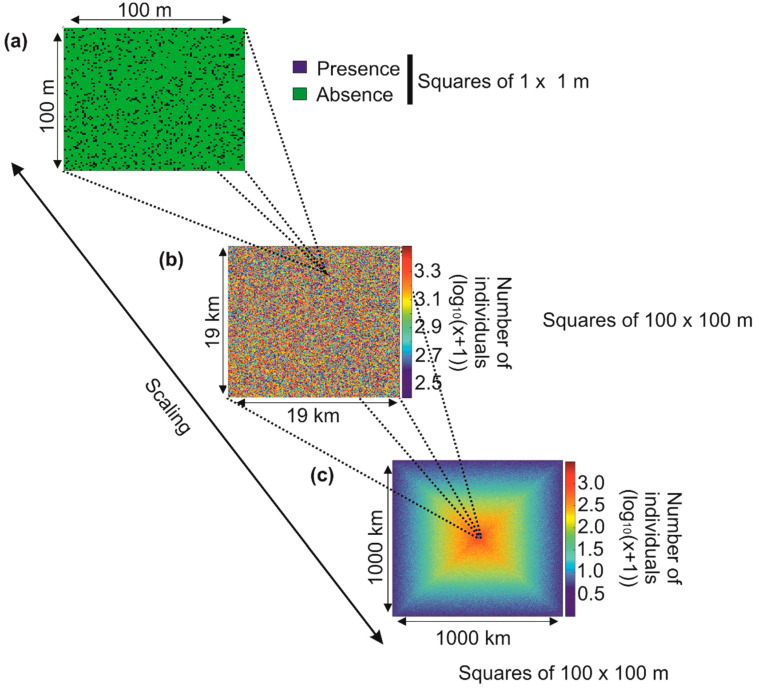
Hypothetical distribution of a species from the scale of a region of 100 × 100 m to a scale of 1000 × 1000 km. (**a**) On a local scale (100 × 100 m), the presence of individuals of the same species (blue squares, 1 × 1 m square) seems random. (**b**) On a more regional scale (19 × 19 km), the number of individuals is counted in each 100 × 100 m square. The density of individuals in the target region still seems random, although this density is between 2.4 and 3.5 (in decimal logarithmic scale). (**c**) On a large scale (1000 × 1000 km), a pattern of variability is clearly observed and the abundance of the species is greater towards the centre of the geographical domain. The transition from small to large scale is called scaling.

**Figure 3 biology-12-00339-f003:**
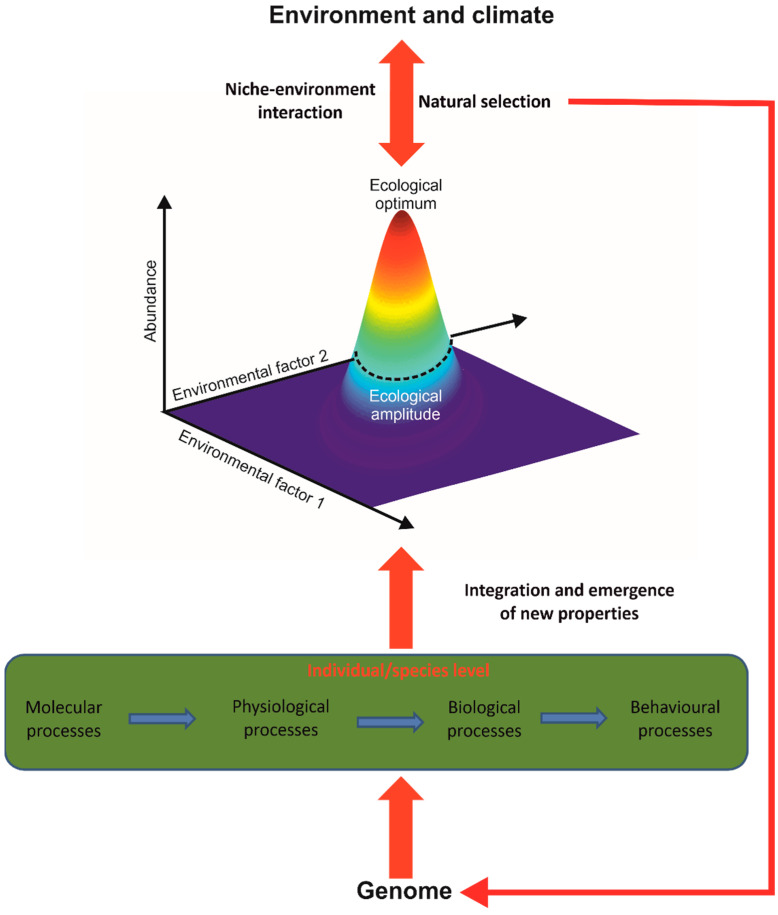
The concept of the ecological niche, the elementary macroscopic brick of METAL. The ecological niche of a species is quantified by simultaneously considering all the ecological factors that influence its abundance. The concept is therefore multidimensional. The ecological optimum represents the values of the ecological parameters for which the maximum abundance is observed. Ecological amplitude is the degree of ecological valence that a species tolerates. Put simply, it is the width of the ecological niche. The use of the ecological niche within METAL makes it possible to integrate molecular, physiological, biological and behavioural processes controlled in part by the genome and the environment. Such processes are impossible to model for all living species on our planet using a reductionist approach. Moreover, the concept of niche makes it possible to consider the emergence of new properties at a specific organisational level. The niche–environment (including climatic) interaction makes it possible to explain, unify and predict a large number of patterns observed in ecology, paleoecology, biogeography and climate change biology. The niche–environment interaction affects the species genome through processes involved in natural selection.

**Figure 4 biology-12-00339-f004:**
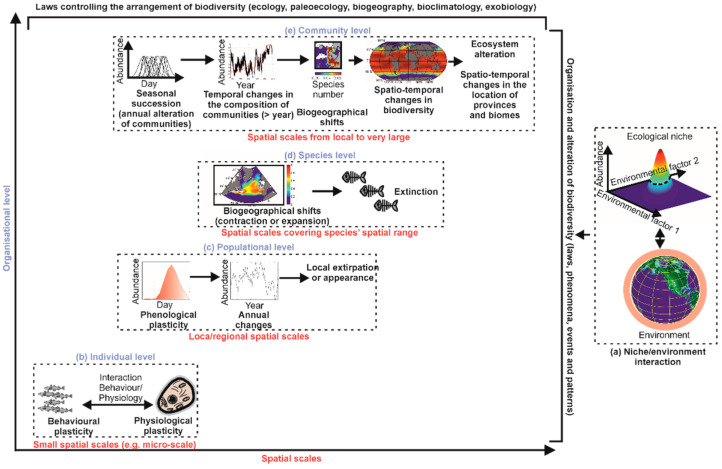
The niche–environment interaction and its influences on the arrangement of biological systems from the individual to the community organisational level and from the micro-scale to the mega-scale. (**a**) Niche–environment interaction. Organisational levels of individual (**b**), population (**c**), species (**d**) and community (**e**). Since the arrangement of communities affects the environment of their habitat, the influence of the niche–environment interaction on the community is also exerted on ecosystems and ecotones, provinces and biomes. In black (bold): phenomena, patterns of variability and biological events. Only the main ones are represented here. In blue (bold): organisational level. In red (bold): spatial scales.

**Figure 5 biology-12-00339-f005:**
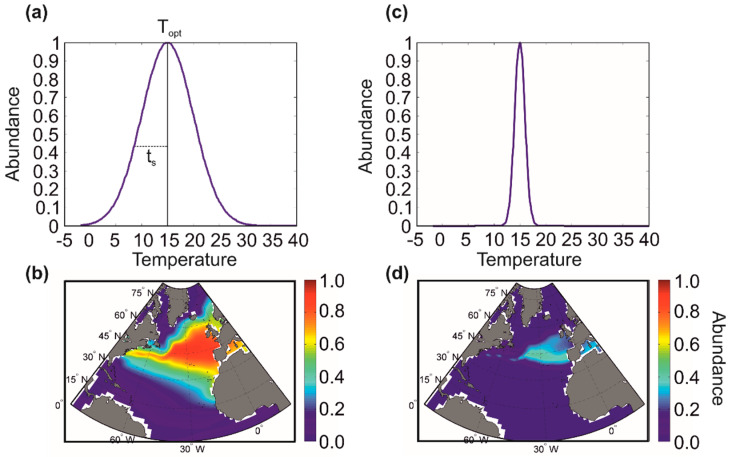
Idealised relationship between the ecological niche of a marine species and its spatial distribution. In this example, the ecological niche is a thermal niche with a Gaussian distribution characterised by two parameters: the optimum temperature and the thermal amplitude (parameter close to the standard deviation). The optimum temperature (T_opt_) is 15 °C for the two fictitious niches (**a**,**c**). The thermal amplitude (t_s_) is higher for (**a**) than (**c**). The spatial distribution is wider and the abundance of the species higher when the species has a thermal niche with a large thermal amplitude (**b**,**d**). In reality, the niche of a species is multidimensional. From Beaugrand and colleagues [23].

**Figure 6 biology-12-00339-f006:**
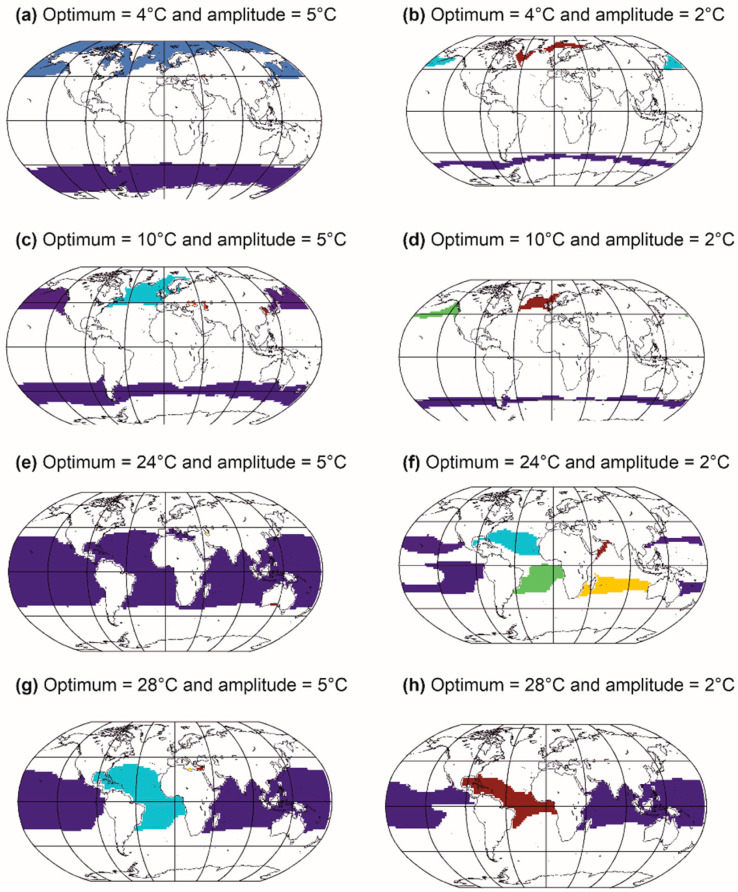
Different types of spatial distribution of marine species generated from thermal niches by varying the thermal optimum and amplitude. The different colours on the map represent different species generated from the same thermal niche. The same niche can give rise to several species if and only if individuals from different species cannot meet (allopatric speciation). Niches with a low thermal amplitude generate more species (e.g., (**a**,**b**) and (**e**,**f**)). The current location of continents at the equator and in the northern latitudes allows more species to form by allopatric speciation. Methods, from Beaugrand and colleagues [29].

**Figure 7 biology-12-00339-f007:**
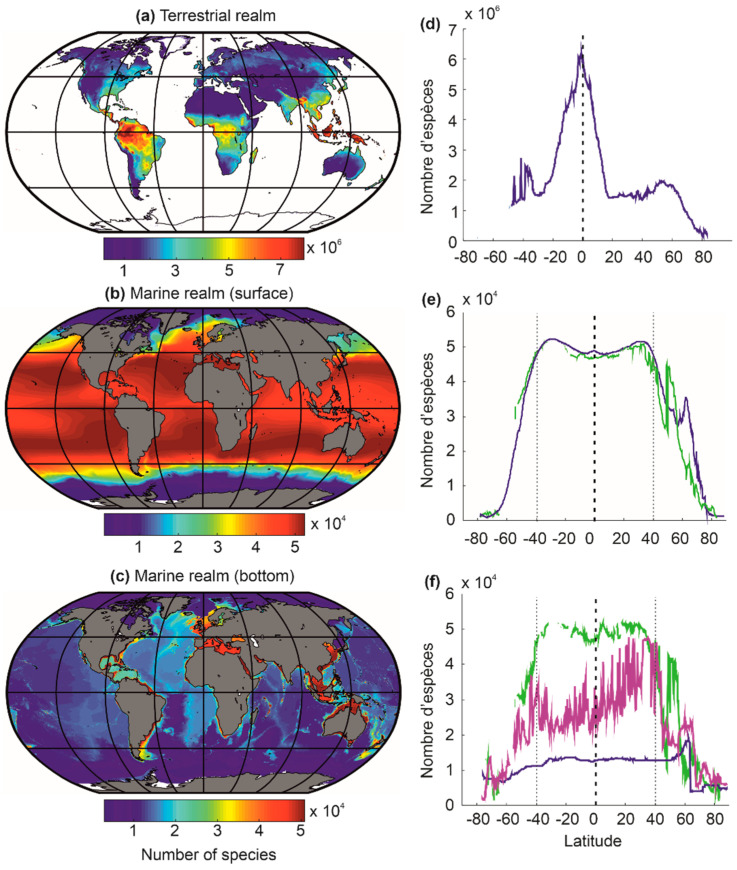
Average distribution of biodiversity (i.e., number of species) in terrestrial (**a**,**d**) and marine (**b**,**e**) surface biodiversity and (**c**,**f**) benthic biodiversity reconstituted from a bioclimatic model derived from METAL [29,100]. (**d**–**f**) The curves show the latitudinal gradient of biodiversity observed for each environment. (**e**) The blue curve reflects the latitudinal biodiversity of the oceanic regions (bathymetry above 200 m) and the green curve reflects the latitudinal biodiversity of the continental-shelf regions (bathymetry below 200 m). (**f**) The curve in green reflects the latitudinal biodiversity of the continental shelf (bathymetry lower than 200 m), the curve in blue reflects that of the deep regions (bathymetry higher than 2000 m), and that in magenta reflects the latitudinal biodiversity of the continental slope (bathymetry between 200 and 2000 m). From Beaugrand and colleagues [29].

**Figure 8 biology-12-00339-f008:**
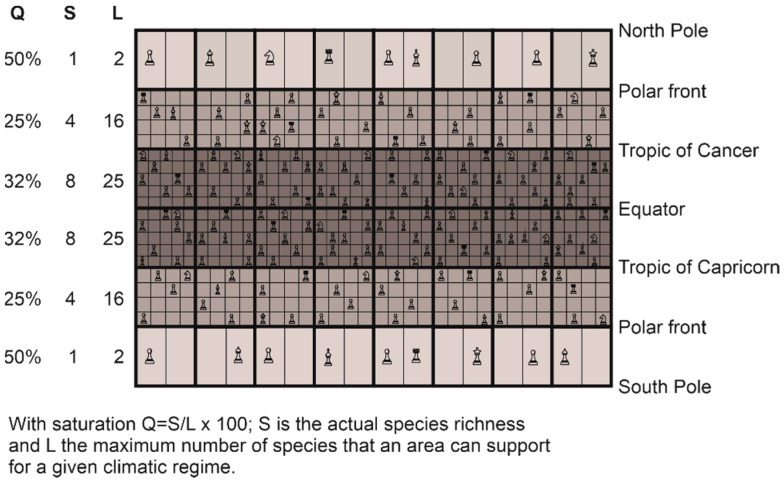
The great chessboard of life that illustrates the mathematical influence on current large-scale biodiversity patterns in the marine realm. Each square on the chessboard, which represents a region, is composed of sub-squares, which represent the number of climatic niches that determines the maximum number of species that can colonise a square (i.e., a region). The different pieces on the chessboard (e.g., king, queen, pawn) symbolize the different biological properties of the species (e.g., their differences in terms of life history traits, such as reproduction). Note that it is also applicable on land. From Beaugrand and colleagues [30].

**Figure 9 biology-12-00339-f009:**
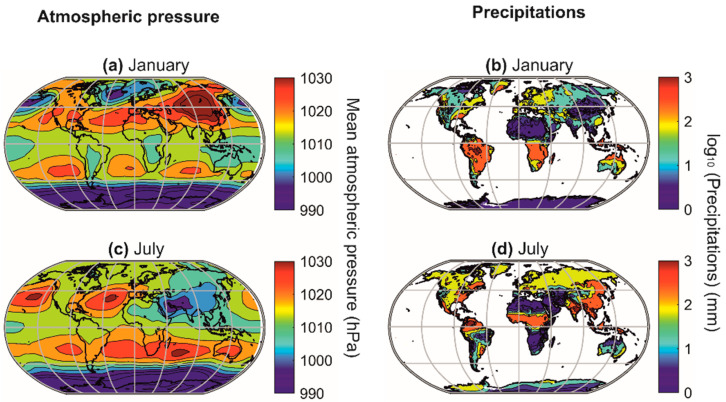
Average positions of the planet’s major pressure highs and lows and their influences on average precipitation in January (**a**,**b**) and July (**c**,**d**). Atmospheric pressure (in hPa) (**a**,**c**). Precipitation (in mm) (**b**,**d**). From Beaugrand [17].

**Figure 10 biology-12-00339-f010:**
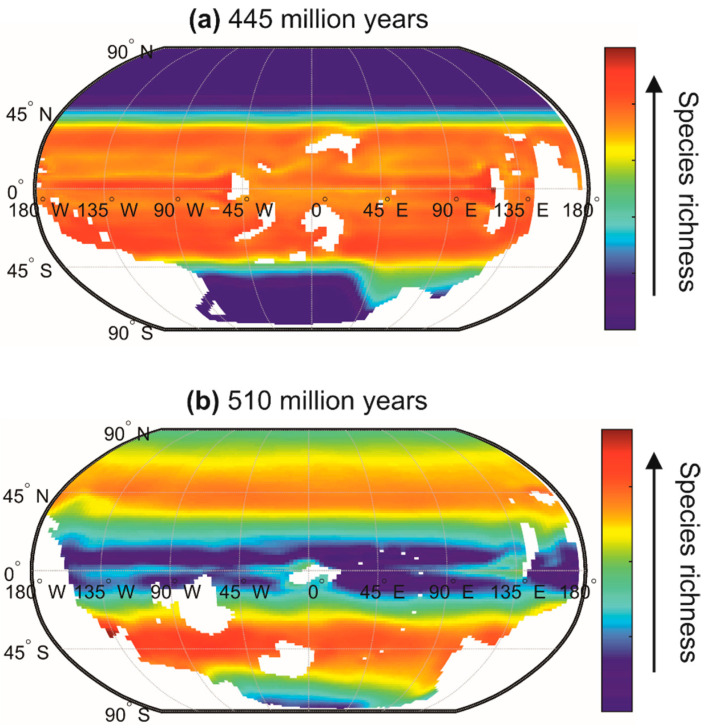
Expected marine surface biodiversity patterns using METAL for (**a**) the end of the Ordovician (445 million years) and (**b**) a warm phase of the Cambrian (510 million years). The position of the continents is indicated in white. Species richness is in relative unit with high richness values in red and low in blue. The conquest of the continents by the first terrestrial plants probably began around 500 million years ago. The carbon dioxide concentration was 5 times the pre-industrial concentration for the late Ordovician and 32 times the pre-industrial concentration for the Cambrian Stage 4. Modified, from Zacaï and colleagues [56].

## Data Availability

The main data used in this paper are available from the corresponding author on reasonable request.

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
