# Peer review of "Towards an Understanding of Large-Scale Biodiversity Patterns on Land and in the Sea"

_biology, 2023, doi:10.3390/biology12030339_

Round 1

Reviewer 1 Report

This review article is a synthesis of several articles written by the author on a theory of ecology, METAL, that addresses a large number of theoretical questions about diversity and biogeography, in terrestrial and marine environments, at different scales of biology. The core of this theory is the Hutchinson niche concept. In particular, the METAL theory proposes to exploit the "interactions" between "niche" and environment. This allows to infer the structure of biodiversity on past or future climate scenarios.

A remarkable result put forward is the observation that the spatial distribution of temperature range, combined with precipitation and seasonal variability, allows to find the major patterns of biogeographic structuring and diversity.

This theory, which corresponds mainly to the integration of certain elements of temporal variability in the parameters of niche definition, has already been published in several articles, so it is not necessary to comment on it in detail here.

However, it is often difficult to distinguish whether the information is new or already published, and an effort of clarification is necessary. Some of the figures, maybe all, are taken from other articles, and this should be mentioned.

The literature used on marine diversity is old, several studies on a global scale have recently published unified results, notably on the latitudinal gradient of diversity in the oceans, on the roles of currents, on the biogeography of niches etc...

I have identified three points that I think are important to consider.

- I do not understand the term "mathematical constraints", it seems to me totally inappropriate. On the other hand I would agree with "physical constraints". Indeed the whole review presents the spatial distribution of niches defined by geophysical parameters (temperature, precipitation, atmospheric pressure). Each place on the planet is characterized by a combination of physical parameters, more or less variable throughout the year, thus defining a type of niche and thus a type of diversity. Geographical distances also come into play in speciation mechanisms.  Diversity is therefore the result of "physical" constraints. Even if mathematics controls physics, it does not come into play here, at least not more than in any biological event. 

- Species principle. Since the author exploits Hutchinson's concept of niche, which has been defined for species, it is surprising that the author does not discuss the evolution of the concept of species in biology. Does the METAL theory remain applicable for microorganisms, prokaryotes, eukaryotes, where the species limit is often blurred?

- Seascape. Concerning marine diversity, it is surprising that the author does not distinguish between plankton and nekton, and does not mention the concept of seascape. This concept defines the fact that it is essential to consider the roles of marine currents (in particular for plankton), which bring constraints on marine ecology that make it very different from terrestrial ecology. Many recent papers have shown that biogeographic structuring results from a spatial structuring of physical parameters, including temperature, created by marine currents. Hellweger et al even demonstrated that biogeographic structuring can result from the simple combination of ocean currents and neutral genetic drift, without any evolutionary adaptation mechanism (Science 345 (6202), 1346-1349 ) .

Minor points:

- Surprisingly, the suitability envelope principle is not mentioned.

- Line 9. Reference?

- Line 42. "extraodinary". This is totally subjective, this adverb is unnecessary.

- line 44 and 47: "inventiveness". This kind of term must be avoided at all costs, as it can evoke "intelligent design". The Oxford dictionary definition: "the ability to THINK of new and interesting ideas". I think the author just means the notion of diversification here.

- Box Text 1. 5 points in each box makes it look like they match one to one but they don't.

- Figure 1: What is the source of this data? This question is valid for all figures.

- line 237. "are due to". "are related to" instead.

- line 274-275. This is totally obvious.

- Line 276. The cause and effect relationship is not at all clear. Better explain. 

- Figure 7. The Y-axes are in French.

- Line 411 - 415. Not at all convincing, see comment above on seascape.

- Line 425. I do not agree with the principle of interaction between environment and niche. Environmental parameters, including their seasonal variability, are part of the niche. I do not see a major increment to the niche concept.

Author Response

Reviewer 1

Reviewer 1 said “This review article is a synthesis of several articles written by the author on a theory of ecology, METAL, that addresses a large number of theoretical questions about diversity and biogeography, in terrestrial and marine environments, at different scales of biology. The core of this theory is the Hutchinson niche concept. In particular, the METAL theory proposes to exploit the "interactions" between "niche" and environment. This allows to infer the structure of biodiversity on past or future climate scenarios.”

Yes this is an excellent summary of the paper.

Reviewer 1 said “A remarkable result put forward is the observation that the spatial distribution of temperature range, combined with precipitation and seasonal variability, allows to find the major patterns of biogeographic structuring and diversity.”

Thank you.

Reviewer 1 said “This theory, which corresponds mainly to the integration of certain elements of temporal variability in the parameters of niche definition, has already been published in several articles, so it is not necessary to comment on it in detail here.”

I concur.

Reviewer 1 said “However, it is often difficult to distinguish whether the information is new or already published, and an effort of clarification is necessary. Some of the figures, maybe all, are taken from other articles, and this should be mentioned.”

I fully agree. I have clarified this point in the revision. At the end of all figures, I refer to the original paper. However, some figures have been specifically designed for this article but those figures (Figures 3 and 4), but one (Figure 2), are schematics. It is also mentioned in the text of the revision. Figure 2 is now better explained in the revised ms.

Reviewer 1 said “The literature used on marine diversity is old, several studies on a global scale have recently published unified results, notably on the latitudinal gradient of diversity in the oceans, on the roles of currents, on the biogeography of niches etc...”

I now mention more recent papers (see the references). I have written a brief overview of the different hypotheses or theories proposed to explain large-scale biodiversity patterns. As a result, the number of references has grown substantially from 62 to 162.

Reviewer 1 said “I have identified three points that I think are important to consider.- I do not understand the term "mathematical constraints", it seems to me totally inappropriate. On the other hand I would agree with "physical constraints". Indeed the whole review presents the spatial distribution of niches defined by geophysical parameters (temperature, precipitation, atmospheric pressure). Each place on the planet is characterized by a combination of physical parameters, more or less variable throughout the year, thus defining a type of niche and thus a type of diversity. Geographical distances also come into play in speciation mechanisms.  Diversity is therefore the result of "physical" constraints. Even if mathematics controls physics, it does not come into play here, at least not more than in any biological event.”

I prefer to keep “mathematical constraints”. I have improved sections 6-9 well and I think the reader can understand better why the niche-environment generates a mathematical constraint on the arrangement of biodiversity. See the following paragraphs page 16 lines 490-540:

“The biogeographical constraints (i.e. limited and elevated number of niches in high and low latitudes, respectively) imposed by the chessboard on biodiversity may be quickly detectable because clade diversification takes place relatively rapidly at a geological time scale [1]. Moreover, in the marine environment, taxa such as plankton have high dispersal capabilities and may rapidly conform to the chessboard. Niche saturation (i.e. the number of observed species on the theoretical number of available niches) may help measuring the degree of conformity of the different taxonomic groups on the chessboard [2]. Although it is commonly assumed that niche saturation increases towards the equator (i) because evolutionary rates are thought to increase from cold to warm regions [3,4] and (ii) because of the presence of strong climate-induced environmental perturbations in extra-tropical regions that limit species richness [5,6], niche saturation is frequently highest towards the poles [2]. These apparently counterintuitive results suggest that the few sub-squares (i.e. the climatic niches) available on the chessboard in polar regions are frequently occupied (Figure 8). This means that low polar biodiversity should not always be attributed to low diversification rates (origination minus extinction)[7,8], but rather to smaller maximum number of species’ niches at saturation that locally limits biodiversity. This low number of niches over polar regions, and inversely the high number of niches equatorwards, originating from the niche-environment interaction, represent a mathematical constraint on the arrangement of biodiversity. Although there remains a great degree of freedom on the type and number of species that can establish into a region (e.g. origination and diversification of a clade), this number cannot exceed a threshold set by the niche-climate interaction.”

“In the marine realm, patterns of niche saturation differ among taxonomic groups, which suggest the existence of a particular chessboard for each group that might originate from taxon-specific diversification history [9,10]. For example, pinnipeds, which exhibit an inversed latitudinal biodiversity gradient, originate from Arctoid carnivores 25–27 Ma in the cold regions of the North Pacific [11]. Diversification processes (e.g. place of origination and time of emergence) may therefore blur large-scale biodiversity patterns imposed by the great chessboard of life. Moreover, life history traits of each group make the great chessboard of life specific to a given taxonomic group, which sometimes explains the lack of universality of biodiversity patterns at large spatial scales (Figure 8) [2].”

“The rate of diversification remains an important parameter because it determines the degree of niche occupation in a given geographical cell. Indeed, when polar regions are excluded, niche saturation of most groups (e.g. plankton and fish) but mammals is higher over permanently stratified regions [12]. Moreover, many clades should exhibit latitudinal biodiversity gradients towards the equator because their probability of emergence should be higher in the tropics where there are more available niches and palaeontological data have provided compelling evidence of greater rates of origination for tropical clades [13]; the hypothesis of Tropical Niche Conservatism [14].”

I also enclose the following paragraph page 20 line 677-688:

“To conclude on this part, there are more terrestrial than marine species because there are more available climatic niches on land. The addition of the water availability dimension in the terrestrial realm (in addition to temperature) morcellates species spatial distribution and increases the influence of landscape heterogeneity and the possibility for allopatric speciation [15]. In the ocean, the seascape is more uniform because there is only a single climate dimension (temperature); see however ref. [16] for other important environmental dimensions. This influence is more prominent in the pelagic than the benthic environment so that it probably explains why there are more benthic than pelagic species [15]. The influence of seascape heterogeneity strongly affects locally biodiversity over seamounts and shelf-edges [17]. Therefore, local biodiversity should be higher over these areas, including heterogeneous shallow ones [15].”

Reviewer 1 said “- Species principle. Since the author exploits Hutchinson's concept of niche, which has been defined for species, it is surprising that the author does not discuss the evolution of the concept of species in biology. Does the METAL theory remain applicable for microorganisms, prokaryotes, eukaryotes, where the species limit is often blurred?”

The METAL theory has been only tested on eukaryotes for which the species concept is clearer. I have added a few sentences in the revision page 3 lines 86-90:

“METAL has not been tested on prokaryotes (Bacteria and Archaea) yet because the species concept is fuzzy in this group, being replaced by Operational Taxonomic Unit (i.e. taxa defined by molecular data analysis) [18,19]. Moreover, the ecological niche of prokaryotes can be more diverse and extreme, especially for Archaea [20,21] and their geographical ranges can be wide [22]. Therefore, all ecological principles examined in this review are only relevant for eukaryotes.”

Reviewer 1 said “- Seascape. Concerning marine diversity, it is surprising that the author does not distinguish between plankton and nekton, and does not mention the concept of seascape. This concept defines the fact that it is essential to consider the roles of marine currents (in particular for plankton), which bring constraints on marine ecology that make it very different from terrestrial ecology. Many recent papers have shown that biogeographic structuring results from a spatial structuring of physical parameters, including temperature, created by marine currents. Hellweger et al even demonstrated that biogeographic structuring can result from the simple combination of ocean currents and neutral genetic drift, without any evolutionary adaptation mechanism (Science 345 (6202), 1346-1349 )”.

I agree. I discuss this point in the revised manuscript page 20 lines 677-686. I say:

“To conclude on this part, there are more terrestrial than marine species because there are more available climatic niches on land. The addition of the water availability dimension in the terrestrial realm (in addition to temperature) morcellates species spatial distribution and increases the influence of landscape heterogeneity and the possibility for allopatric speciation [15]. In the ocean, the seascape is more uniform because there is only a single climate dimension (temperature); see however ref. [16] for other important environmental dimensions. This influence is more prominent in the pelagic than the benthic environment so that it probably explains why there are more benthic than pelagic species [15]. The influence of seascape heterogeneity strongly affects locally biodiversity over seamounts and shelf-edges [17]. Therefore, local biodiversity should be higher over these areas, including heterogeneous shallow ones [15].”

I have also added the following sentence Page 17 lines 560-562:

Note that it is possible to do more specific simulations to account for the biology of a specific clade such as plankton, fish, coral reef or mangroves [15,23].

The important reference has also been included in the revision.

Reviewer 1 said “- Surprisingly, the suitability envelope principle is not mentioned.”

I have added a few sentences in the text page 7 lines 211-216:

“Also known as species distribution models (SDMs) or bioclimatic envelope models [24,25], METAL integrates ecological niche models (ENMs) in its framework. ENMs primarily assessed the realised niche (not the fundamental niche) based on past or contemporary species spatial distribution and the knowledge of some environmental variables. They then used the realised niche to project the likely distribution of a species in the past, present or future. ENMs have been extensively applied to project future species spatial distributions in the context of global climate change [26-29].”

Reviewer 1 said “- Line 9. Reference?”

I cannot add any reference in the abstract from my understanding of the instruction to authors. The references are included in Section 5.1, second sentence. I say:

Among scientific questions such as the origin of life, the biological basis of consciousness or the composition of the universe, this question was cited as one of the 25 most important enigmas by the American journal Science in 2005 [30,31].”

Reviewer 1 said “- Line 42. "extraodinary". This is totally subjective, this adverb is unnecessary.”

The word has been removed from the revision.

Reviewer 1 said “- line 44 and 47: "inventiveness". This kind of term must be avoided at all costs, as it can evoke "intelligent design". The Oxford dictionary definition: "the ability to THINK of new and interesting ideas". I think the author just means the notion of diversification here.”

I fully agree. I have removed the word in the revision. It has been replaced by “diversity” (see lines 46 and 48 page 2).

Reviewer 1 said “- Box Text 1. 5 points in each box makes it look like they match one to one but they don't.”

I have removed the numbers by crosses (See Text Box 1).

Reviewer 1 said “- Figure 1: What is the source of this data? This question is valid for all figures.”

This figure has been modified from Beaugrand and colleagues [32]. Figures 2 has been specifically designed for this review. The next two schematics have also been designed for the present review. The others are from other papers. The references have been included in all figure legends.

Reviewer 1 said “- line 237. "are due to". "are related to" instead.”

This has been replaced in the revision.

Reviewer 1 said “- line 274-275. This is totally obvious.”

The paragraph has been clarified. I say page 12 lines 377-391:

“We can thus create a multitude of pseudospecies by varying the optimum and the ecological amplitude of each niche dimension. Figure 6 shows the creation of marine pseudospecies from a simple Gaussian thermal niche  [33,34]. Different thermal optima and amplitudes are used [33,34]. In this example, when distributional ranges originating from one thermal niche are spatially separated, it is considered that they represent different species; therefore one niche can give several species, in agreement with Buffon’s Law also known as the first principle of biogeography [35]. We see that thermal niches with lower thermal amplitudes give more species but these are characterized by smaller distributional ranges (Figure 6, left maps versus right maps). Figure 5 shows that there is a relationship between the average abundance of a species and its area of distribution, a relationship already demonstrated empirically by Brown [36]. We extended this relationship by indicating that there is a positive link between the ecological amplitude of a species, its average abundance and its distribution area [34] (Figures 5 and 6).

Reviewer 1 said “- Line 276. The cause and effect relationship is not at all clear. Better explain.” 

This part has been improved. I say page 12 lines 373-377:

“Figure 5 shows that there is a relationship between the average abundance of a species and its area of distribution, a relationship already demonstrated empirically by Brown [36]. We extended this relationship by indicating that there is a positive link between the ecological amplitude of a species, its average abundance and its distribution area [34] (Figures 5 and 6)”.

Reviewer 1 said “- Figure 7. The Y-axes are in French.”

Figure 7 has been modified in the revision.

Reviewer 1 said “- Line 411 - 415. Not at all convincing, see comment above on seascape.”

Reviewer 1 said “- Line 425. I do not agree with the principle of interaction between environment and niche. Environmental parameters, including their seasonal variability, are part of the niche. I do not see a major increment to the niche concept.”

This was not clear. I agree that seasonal variability is indeed included in the niche concept. I have clarified this point page 6, line 171-182. I say:

“ METAL considers the fundamental niche (i.e. the niche without the influence of species interaction) and current models do not explicitly include the influence of biotic interaction yet [15,37]. The niche can be divided into five components: (i) climatic, (ii) physico-chemical, (iii) substrate, or trophic with (iv) dietary and (v) resource concentration components [34,37,38]. It integrates all environmental conditions where a species’ individual can ensure its homeostasis, grow and reproduce. A species’ niche therefore includes phenotypic plasticity, encompassing polyphenism and reaction norm (i.e. a species niche integrates the niches of all individuals of that species).”

References added in my answers

  1. Rabosky, D.L.; Hurlbert, A.H. Species richness at continental sclaes is dominated by ecological limits. The American Naturalist 2015 185, 572-583.
  2. Beaugrand, G.; Luczak, C.; Goberville, E.; Kirby, R.R. Marine biodiversity and the chessboard of life Plos One 2018, 13, e0194006, doi:https://doi.org/10.1371/journal.pone.0194006.
  3. Gillooly, J.; Allen, A.P.; West, G.B.; Brown, J.H. The rate of DNA evolution: effects of body size and temperature on the molecular clock. Proceedings of the National Academy of Sciences of the United States of America 2005, 102, 140-145.
  4. Machac, A.; Zrzavy, J.; Smrckova, J.; Storch, D. Temperature dependence of evolutionary diversification: differences between two contrasting model taxa support the metabolic theory of ecology. Journal of Evolutionary Biology 2012, 25, 2449-2456.
  5. Crame, J.A. Pattern and proceses in marine biogeography: a view from the poles. In Fontiers of biogeography I: new directions in the geography of nature, Lomolino, M.V., Heaney, L.R., Eds.; MA: Sinauer Associates: Sunderland, 2004; pp. 272-292.
  6. Rohde, K. Nonequilibrium ecology; Cambridge University Press: Cambridge, 2005.
  7. Jablonski, D.; Roy, K.; Valentine, J.W. Out of the Tropics: evolutionary dynamics of the latitudinal diversity gradient. Science 2006, 314, 102-106.
  8. Dowle, E.J.; Morgan-Richards, M.; Trewick, S.A. Molecular evolution and the latitudinal biodiversity gradient. Heredity 2013, 110, 501–510, doi:10.1038/hdy.2013.4.
  9. Benton, M.J. Diversification and extinction in the history of life. Science 1995, 268, 52-58.
  10. Morlon, H. Phylogenetic approaches for studying diversification. Ecology Letters 2014, 17, 508–525.
  11. Berta, A.; Adam, P. The evolutionary biology of pinnipeds. In Secondary adaptation of tetrapods to life in the water de Buffrenil, V., Mazin, J.-M., Eds.; Verlag Dr Frederich Pfeil: Munchen Germany, 2001; pp. 235-260.
  12. Peters, R.H. The ecological implications of body size; Cambridge University Press: Cambridge, 1983; p. 329.
  13. Wiens, J.J.; Donoghue, M.J. Historical biogeography, ecology and species richness. Trends in Ecology and Evolution 2004, 19, 639-644.
  14. Crisp, M.D.; Cook, L.G. Phylogenetic niche conservatism: what are the underlying evolutionary and ecological causes? New Phytologist 2012, 196, 681-694.
  15. Beaugrand, G.; Kirby, R.R.; Goberville, E. The mathematical influence on global patterns of biodiversity. Ecology and Evolution 2020, 10, 6494-6511, doi: 10.1002/ece3.6385.
  16. Kléparski, L.; Beaugrand, G.; Kirby, R.R. How do plankton species coexist in an apparently unstructured environment? Biology Letters 2022, 18, 20220207, doi:https://doi.org/10.1098/rsbl.2022.0207.
  17. Swanborn, D.J.B.; Huvenne, V.A.I.; Pittman, S.J.; Rogers, A.D.; Taylor, M.L.; Woodall, L.C. Mapping, quantifying and comparing seascape heterogeneity of Southwest Indian Ridge seamounts. Landscape Ecology 2023, 38, 185-203.
  18. Hellweger, F.L.; Van Sebille, E.; Fredrick, N.D. Biogeographic patterns in ocean microbes emerge in a neutral agent-based model. Science 2014, 345, 1346-1349.
  19. Lladó Fernández, S.; Větrovský, T.; Baldrian, P. The concept of operational taxonomic units revisited: genomes of bacteria that are regarded as closely related are often highly dissimilar. Folia Microbiologica 2018, 64, 19-23.
  20. Cohan, F.M.; Koeppel, A.F. The origins of ecological diversity in prokaryotes. Current Biology 2008, 18, R1024-R1034, doi:10.1016/j.cub.2008.09.014.
  21. Suen, G.; Goldman, B.S.; Welch, R.D. Predicting prokaryotic eological niches using genome sequence analysis. PLoS ONE 2007, 2, e743, doi:10.1371/journal.pone.0000743.
  22. Hughes Martiny, J.B.; Bohannan, B.J.M.; Brown, J.H.; Colwell, R.K.; Fuhrman, J.A.; Green, J.L.; Horner-Devine, M.C.; Kane, M.; Krumins, J.A.; Kuske, C.R.; et al. Microbial biogeography: putting microorganisms on the map. Nature Reviews Microbiology 2006, 4, 102–112.
  23. Speck, L. Caractérisation de la biodiversité marine à l’aide de la théorie METAL; Université du Littoral Côte d'Opale: Wimereux, 2022; p. 22.
  24. Araújo, M.B.; Peterson, A.T. Uses and misuses of bioclimatic envelope modelling. Ecology 2012, 93, 1527-1539, doi:org/10.1890/11-1930.1.
  25. Thuiller, W.; Lafourcade, B.; Engler, R.; Araújo, M.B. BIOMOD - A platform for ensemble forecasting of species distributions. Ecography 2009, 32, 369-373, doi:10.1111/j.1600-0587.2008.05742.x.
  26. Goberville, E.; Beaugrand, G.; Hautekeete, N.-C.; Piquot, Y.; Luczak, C. Uncertainties in species distribution projections and general circulation models. Ecology and Evolution 2015, 5, 1100-1116, doi:10.1002/ece3.1411.
  27. Raybaud, V.; Beaugrand, G.; Goberville, E.; Delebecq, G.; Destombe, C.; Valero, M.; Davoult, D.; Morin, P.; Gevaert, F. Decline in Kelp in West Europe and Climate. PLOS One 2013, 8, e66044, doi:10.1371/journal.pone.0066044.
  28. Rombouts, I.; Beaugrand, G.; Dauvin, J.-C. Potential changes in benthic macrofaunal distributions from the English Channel simulated under climate change scenarios. Estuarine, coastal and Shelf Science 2012, 99, 153-161.
  29. Schickele, A.; Goberville, E.; Leroy, B.; Beaugrand, G.; Hattab, T.; Francour, P.; Raybaud, V. Redistribution of small pelagic fish in Europe and Climate Change. Fish and Fisheries 2021, 22, 212-225.
  30. Kennedy, D.; Norman, C. What don't we know? Science 2005, 309, 75.
  31. Pennisi, E. What determines species diversity? Science 2005, 309, 90.
  32. Beaugrand, G.; Ibañez, F.; Lindley, J.A. An overview of statistical methods applied to the CPR data. Progress in Oceanography 2003, 58, 235-262.
  33. Beaugrand, G. Theoretical basis for predicting climate-induced abrupt shifts in the oceans. Philosophical Tansactions of the Royal Society B: Biological Sciences 2015, 370 20130264, doi:10.1098/rstb.2013.0264.
  34. Beaugrand, G.; Goberville, E.; Luczak, C.; Kirby, R.R. Marine biological shifts and climate. Proceedings of the Royal Society B: Biological Sciences 2014, 281, 20133350, doi:10.1098/rspb.2013.3350.
  35. Lomolino, M.V.; Riddle, B.R.; Brown, J.H. Biogeography, 3 ed.; Sinauer Associates, Inc.: Sunderland, 2006; p. 845.
  36. Brown, J.H. On the relationship between abundance and distribution of species. The American Naturalist 1984, 124, 255-279.
  37. Beaugrand, G.; Balembois, A.; Kléparski, L.; Kirby, R.R. Addressing the dichotomy of fishing and climate in fishery management with the FishClim model. Communications Biology 2022, 5, 1146, doi:10.1038/s42003-022-04100-6.
  38. Caracciolo, M.; Beaugrand, G.; Hélaouët, P.; Gevaert, F.; Edwards, M.; Lizon, F.; Kléparski, L.; Goberville, E. Annual phytoplankton succession results from niche-environment interaction. Journal of Plankton Research 2021, 43, 85-102, doi:10.1093/plankt/fbaa060.

Reviewer 2 Report

The main reasons for rejecting this manuscript are:

             1)       The English is not sufficient for the peer review process;

2)       The figures are not complete or are not clear enough to read:

3)       References are incomplete or very old;

4)       The article contains observations but is not a full study;

5)       It discusses findings in relation to some of the work in the field but ignores other important work;

6)       The study lacked clear control groups or other comparison metrics;

7)       The study did not conform to recognized procedures or methodology that can be repeated;

8)       The analysis is not statistically valid or does not follow the norms of the field;

9)       The arguments are illogical, unstructured or invalid;

10)   The data does not support the conclusions;

11)   The conclusions ignore large portions of the literature.

Author Response

Reviewer 2

Reviewer 2 said “The main reasons for rejecting this manuscript are:”

Reviewer 2 lists 11 reasons why he/she recommends rejection of the ms. I am answering to all points raised by the reviewer.

Reviewer 2 said “The English is not sufficient for the peer review process”

The English has been improved with the help of a native English speaker.

Reviewer 2 said “The figures are not complete or are not clear enough to read”.

I have better described the figures in the revised manuscript (see the revision).

Reviewer 2 said “References are incomplete or very old”

We have added 100 references in the revision, many are recent. But I need to keep the old ones.

Reviewer 2 said “The article contains observations but is not a full study”

As said in the first page of the manuscript, the paper is a review.

Reviewer 2 said “It discusses findings in relation to some of the work in the field but ignores other important works”

We have improved the number of references by 100.

Reviewer 2 said “The study lacked clear control groups or other comparison metrics”

I do not understand this comment. It seems unrelated to the current manuscript. I recall here as well that the paper is a review, as clearly indicated in the first page of the manuscript.

Reviewer 2 said “The study did not conform to recognized procedures or methodology that can be repeated”.

All methodology used in the review paper has been published elsewhere (nature climate change, global change biology, proceedings B, Plos One, Ecology and Evolution, Phylosophical transaction of the royal society B, Science Advances). See the references in the ms. I have clarified this point in the revision. 

Reviewer 2 said “The analysis is not statistically valid or does not follow the norms of the field”

In this review paper, there is no new analyses. As I said earlier, l have already published these analyses in other scientific journals. I have clarified this point in the revision.

Reviewer 2 said “The arguments are illogical, unstructured or invalid”

It would be helpful that Reviewer 2 says which arguments are illogical, unstructured or invalid.

Reviewer 2 said “The data does not support the conclusions”

Again, this is a review paper. All results presented in this paper has been published elsewhere.

 Reviewer 2 said “The conclusions ignore large portions of the literature.”

Reviewer 2 is right. I have increased the number of references in the review paper by 100.

Reviewer 3 Report

This manuscript present a new theory (METAL) that proposes that biodiversity across oceans and land is influenced by the climate and the environment in a deterministic manner, based upon the ecological niche concept. In this sense, the manuscript addresses a relevant subject. The main concern is that this theory has been already presented in detail by the same author in Beaugrand et al. 2020. Ecology and Evolution 2020, 10, 6494-6511. Furthermore, Sections 1 (Introduction), 2 and 3 present very general concepts of biology, at least sections 2 and 3 and Box Text 1 can be omitted. The only section I think is original and interesting is section 7 (L355-397) where METAL is applied to Ordovician and Cambrian eras and latitudinal gradients of biodiversity are reverted.

Specific comments:

In introduction, the objective of the study is not presented. The objective is presented at the end of section 4.

Figure 5 and related text is very similar to what is presented in Figure 6, hence it can be omitted.

L276. Average abundance refers to local abundance?

L290. How niches of pseudo-species are defined? randomly assigning temperature optima and range? or based on real species niches?

L304 (Figure 7 legend). "Nombre d'especes". translate to English.

L339-353. In the presented chessboard, Equator have niche % saturation lower than in poles. However, according to observed patterns, I would say that in Equator the niche % saturation should be greater than in poles since the number of species in the similar environment is greater.

L362. "climatic stability and heat increases surface biodiversity in the ocean environment". However, this may apply to benthic but not in the pelagic domain.

L379-389. Very interesting!

L398-415. Section 7 should be section 8. The conundrum of why the number of terrestrial species is higher than in ocean, whilst marine biodiversity appeared long before in land, should cite R. May: May, R. (1994). Biological diversity: differences between land and sea. Philosophical Transactions of the Royal Society of London. Series B: Biological Sciences, 343, 105-111.

L417-420. Unsure that Galileo citation is useful here in the main conclusion of the manuscript

L434-438. To me, METAL does not incorporate evolutionary processes, hence, Dobzansky citation is not clearly fitted here.  

Author Response

Reviewer 3

Reviewer 3 said “ This manuscript presents a new theory (METAL) that proposes that biodiversity across oceans and land is influenced by the climate and the environment in a deterministic manner, based upon the ecological niche concept. In this sense, the manuscript addresses a relevant subject. The main concern is that this theory has been already presented in detail by the same author in Beaugrand et al. 2020. Ecology and Evolution 2020, 10, 6494-6511. Furthermore, Sections 1 (Introduction), 2 and 3 present very general concepts of biology, at least sections 2 and 3 and Box Text 1 can be omitted. The only section I think is original and interesting is section 7 (L355-397) where METAL is applied to Ordovician and Cambrian eras and latitudinal gradients of biodiversity are reverted.”

The objective of this review is to provide a global view of METAL. Each paper I refer to has only tested a part of the theory. That is why I think that all parts are necessary because they provide a full and coherent view of the theory. I hope that the revised manuscript will clarify this point.

Reviewer 3 said “ In introduction, the objective of the study is not presented. The objective is presented at the end of section 4.”

I agree. I have added a paragraph at the end of Section 1 page 2 lines 69-90:

“In this review, I present the MacroEcological Theory on the Arrangement of Life (METAL), a theory that proposes that biodiversity is strongly influenced by the climatic and the environmental regime in a deterministic manner (https://biodiversite.macroecologie.climat.cnrs.fr). This influence mainly occurs through the interactions between the ecological niche of species sensu Hutchinson (i.e. the range of a species tolerance when several factors are taken simultaneously) and the climate and  the environment [1]. The niche-environment interaction is therefore a fundamental interaction in ecology that enables one to predict and unify (i) at a species level local changes in abundance, species phenology, and biogeographic range shifts, and (ii) at a community level, the arrangement of biodiversity in space and time as well as long-term community/ecosystem shifts, including regime shifts [2-11]. This theory offers a way to make testable ecological and biogeographical predictions to understand how life is organized, and how it responds to global environmental changes [12]. More specifically here, I show how METAL helps understanding (i) why there are more species at low latitudes than at the poles, (ii) why the peak of biodiversity is located at mid-latitudes in the oceanic domain and at the equator in the terrestrial domain and (iii) finally why there are more terrestrial than marine species despite the fact that biodiversity has emerged in the oceans. METAL has not been tested on prokaryotes (Bacteria and Archaea) yet because the species concept is fuzzy in this group, being replaced by Operational Taxonomic Unit (i.e. taxa defined by molecular data analysis) [13,14]. Moreover, the ecological niche of prokaryotes can be more diverse and extreme, especially for Archaea [15,16] and their geographical ranges can be wide [17]. Therefore, all ecological principles examined in this review are only relevant for eukaryotes.”

Reviewer 3 said “ Figure 5 and related text is very similar to what is presented in Figure 6, hence it can be omitted.”

I think it is better to keep it because Figure 5 shows the relationship between a Gaussian niche and the abundance within the spatial range of a species while Figure 6 shows how a single niche can lead to several species (Buffon’s Law). I think the text was not clear enough and I have improved this part to make the difference between the two figures understandable. I say page 12 lines 393-419:

“We can thus create a multitude of pseudospecies by varying the optimum and the ecological amplitude of each niche dimension. Figure 6 shows the creation of marine pseudospecies from a simple Gaussian thermal niche  [2,4]. Note that different types of niches can be used from rectangular to trapezoidal [6,18] and from logistic to Beta distribution [6,8], symmetrical or asymmetrical [6], parametric or nonparametric [4]. Moreover, the niche can be fully multidimensional [19], including nutrients, solar radiation or mixed layer depth for phytoplankton, bathymetry and sediment type for fish, soil pH and composition for plants [6,19-23]. So far, most METAL simulations were based on niches that vary between 0 (i.e. absence of a species for a given environmental regime) to 1 (i.e. highest abundance, or presence in case of a rectangular niche). Therefore, all species can reach the same level of maximum abundance. Although this assumption may possibly hold for a clade composed of species with a similar size [24-26], this is not so for a group that exhibits large size variability (e.g. mammals). Note, however, that this assumption does not affect biodiversity when the selected indicator is species richness (see below).”

Different thermal optima and amplitudes are used [2,4]. In this example, when distributional ranges originating from one thermal niche are spatially separated, it is considered that they represent different species; therefore one niche can give several species, in agreement with Buffon’s Law also known as the first principle of biogeography [27]. We see that thermal niches with lower thermal amplitudes give more species but these are characterized by smaller distributional ranges (Figure 6, left maps versus right maps). Figure 5 shows that there is a relationship between the average abundance of a species and its area of distribution, a relationship already demonstrated empirically by Brown [28]. We extended this relationship by indicating that there is a positive link between the ecological amplitude of a species, its average abundance and its distribution area [4] (Figures 5 and 6).”

I have also added the following paragraph page 13 line 428-441:

“Examples from Figure 6 show that a niche can lead to more pseudospecies in the Northern than in the Southern Hemisphere (Figures 6b-d); this is due to the current location of continents that acts as a barrier against gene flux, triggering more allopatric speciation in the Northern than the Southern Hemisphere (towards high latitudes). When the thermal amplitude is larger, a single niche leads to less pseudospecies; e.g. only one in Figure 6a in each hemisphere. Moreover, the current configuration (i.e. south to north configuration) of the continents also enables more pseudospecies to emerge in the tropics (Figures 6g-h), especially when the pseudospecies are stenoecious (Figure 6e versus Figure 6f). Note that parapatric and sympatric speciations are not accounted for in this example. Allopatric speciation is thought to be a widespread mode of speciation in the marine environment, despite more and more evidence that other modes of speciation might also be play a role [29]. Parapatric speciation is thought to be possible in the ocean [30-32]. Clinal parapatric speciation has been suggested for salps and some benthic species [29,33]. Sympatric speciation might also be frequent for marine invertebrates [34].”

Reviewer 3 said “ L276. Average abundance refers to local abundance?”

This is average abundance in this context. The modifications of the text (see above)

Reviewer 3 said “ L290. How niches of pseudo-species are defined? randomly assigning temperature optima and range? or based on real species niches?”

I have clarified this point in the revision pages 13-14 lines 442-461:

“To reproduce the large-scale arrangement of biodiversity, we can build a model that first creates millions of niches, which then allow pseudospecies to establish themselves in a given region so long as environmental fluctuations are suitable [1,10,11,18]. The principle of the model is simple. It starts to create a large number of niches on the basis of temperature only (marine realm) or using both temperature and precipitation (terrestrial realm). The niches, which can overlap, are created (i) for temperature between -1.8°C and 44°C in both realms and (ii) for precipitation between 0 and 3,000 mm; the full procedure is described in ref. [10]. At the end of the procedure, there are 101,397 and 94,299,210 niches in the marine and terrestrial realms, respectively. About 25% and 1% of these niches are chosen randomly to perform the simulations in the marine and terrestrial realms, respectively [10]. The use of fictitious niches and species is especially useful since we have only inventoried 9% of marine and 14% of terrestrial biodiversity and we know little about the biology of most species (see Section 5.1) A niche can give rise to several pseudospecies if individuals from different regions never come into contact (e.g. Figure 6f) [10]. Pseudospecies are gradually colonizing the terrestrial and marine environment (surface and bottom). During the simulations, the species gradually organize themselves into communities and the biodiversity, more precisely here the number of species in a given region, is reproduced.”

Reviewer 3 said “ L304 (Figure 7 legend). "Nombre d'especes". translate to English.”

This has been modified in the revision.

Reviewer 4 said “ L339-353. In the presented chessboard, Equator have niche % saturation lower than in poles. However, according to observed patterns, I would say that in Equator the niche % saturation should be greater than in poles since the number of species in the similar environment is greater.”

There are indeed more saturated niches at the poles because there are less niches for many taxonomic groups (see the figure below). At a first glance, this may seem counterintuitive but this is related to the fact that there are a low number of niches available. I have better explained this in the revision. See pages 16-17 lines 545-600.

These apparently counterintuitive results suggest that the few sub-squares (i.e. the climatic niches) available on the chessboard in polar regions are frequently occupied (Figure 8). This means that low polar biodiversity should not always be attributed to low diversification rates (origination minus extinction)[35,36], but rather to smaller maximum number of species’ niches at saturation that locally limits biodiversity. This low number of niches over polar regions, and inversely the high number of niches equatorwards, originating from the niche-environment interaction, represent a mathematical constraint on the arrangement of biodiversity. Although there remains a great degree of freedom on the type and number of species that can establish into a region  (e.g. origination and diversification of a clade), this number cannot exceed a threshold set by the niche-climate interaction.

In the marine realm, patterns of niche saturation differ among taxonomic groups, which suggest the existence of a particular chessboard for each group that might originate from taxon-specific diversification history [37,38]. For example, pinnipeds, which exhibit an inversed latitudinal biodiversity gradient, originate from Arctoid carnivores 25–27 Ma in the cold regions of the North Pacific [39]. Place of origination and time of emergence may therefore blur large-scale biodiversity patterns imposed by the great chessboard of life. Moreover, life history traits of each group make the great chessboard of life specific to a given taxonomic group, which sometimes explains the lack of universality of biodiversity patterns at large spatial scales (Figure 8) [11].

The rate of diversification remains an important parameter because it determines the degree of niche occupation in a given geographical cell. Indeed, when polar regions are excluded, niche saturation of most groups (e.g. plankton and fish) but mammals is higher over permanently stratified regions [24]. Moreover, many clades should exhibit latitudinal biodiversity gradients towards the equator because their probability of emergence should be higher in the tropics where there are more available niches and palaeontological data have provided compelling evidence of greater rates of origination for tropical clades [40]; the hypothesis of Tropical Niche Conservatism [41].

Beaugrand and colleagues  suggest that the total number of species on the chessboard diminishes with organismal complexity [11], which can be explained by basic ecological and evolutionary processes. Endosomatic energy decreases from primary producers to top predators as a consequence of the second law of thermodynamics, decreasing the number of individuals and thereby species richness and thereby niche saturation from producers to higher trophic levels [1,27]. A positive relationships between number of individuals and species richness has often been proposed to explain large-scale biodiversity patterns; e.g. the productivity theory [42], the Area Hypothesis [43], and the Unified Neutral Theory of Biodiversity and Biogeography [44]. In addition to diminishing the number of individuals [45], larger body size also increases generation time [24], which slows down evolution [45,46]. Therefore, the likelihood a taxon exhibits a large-scale biodiversity pattern different to the one imposed by the great chessboard of life is greater when its mean niche saturation is lower. This is especially the case for marine mammals. Pinnipeds, which have a low degree of niche saturation (<1%, [11]), exhibit a pattern that does not conform to the great chessboard of life [11]. Note that it is possible to do specific simulations to account for the biology of a specific clade such as plankton, fish, coral reef or mangroves [10,20].”

Fig 1. Niche saturation for (A) foraminifers, (B) euphausiids, (C) oceanic sharks, (D) tuna/billfish, (E) cetaceans and (F) pinnipeds. For niche saturation, when values tend to 1, this indicates a large degree of niche saturation, and inversely. Oceanic areas in white are missing data either because of the proximity of a geographical square to the coast or high sea ice concentration. From ref. [11].

Reviewer 4 said “ L362. "climatic stability and heat increases surface biodiversity in the ocean environment". However, this may apply to benthic but not in the pelagic domain.”

I am not sure I understand the comment. I would read the comment as “However, this may apply to pelagic but not benthic in the domain.” If this is what the reviewer means, I agree and this is why large-scale biodiversity patterns are singular at depth. I already discussed that page 18 lines 623-626. I say:

“Finally, the cold ocean floors do not show a typical biodiversity gradient but a very homogeneous biodiversity pattern except in high-latitude regions where biodiversity decreases slightly and over seamounts where it is higher (Figure 7e-f).”

Reviewer 3 said “ L379-389. Very interesting!”

Thank you.

Reviewer 3 said “ L398-415. Section 7 should be section 8. The conundrum of why the number of terrestrial species is higher than in ocean, whilst marine biodiversity appeared long before in land, should cite R. May: May, R. (1994). Biological diversity: differences between land and sea. Philosophical Transactions of the Royal Society of London. Series B: Biological Sciences, 343, 105-111.”

I have changed the number. In addition, I have also mentioned May (1994) in the revision because it very relevant here.

Reviewer 3 said “ L417-420. Unsure that Galileo citation is useful here in the main conclusion of the manuscript”

I would prefer to keep it because the review suggests that there is a strong mathematical constraint to the current arrangement of biodiversity. Referring to Galileo shows that the mathematical influence is not only found in physics but also in biology.

Reviewer 3 said “ L434-438. To me, METAL does not incorporate evolutionary processes, hence, Dobzansky citation is not clearly fitted here.”  

I have clarified this point. I now say page 21 lines 756-766:

“The establishment of a global theory of biodiversity, however, requires taking into account a large number of biological processes that also influence biodiversity (e.g. diversification rate and origination place of a clade) and Theodosius Dobzhansky was greatly inspired when he wrote his article “Nothing in biology makes sense except in the light of evolution” [47]. In addition to other key ecological factors discussed in Section 5, the METAL theory should therefore consider more explicitly some key evolutionary processes in the future. In the process of developing such a global theory of biodiversity, considering all the complexity of biological systems (Text Box 1), it is important to recognize that mathematical constraints caused by (i) the number of key dimensions that the niches include in the terrestrial and marine realms and (ii) the niche-environment interaction also control the arrangement of biodiversity.”

References added in my answers

  1. Beaugrand, G. Marine biodiversity, climatic variability and global change.; Routledge: London, 2015; p. 474.
  2. Beaugrand, G. Theoretical basis for predicting climate-induced abrupt shifts in the oceans. Philosophical Tansactions of the Royal Society B: Biological Sciences 2015, 370 20130264, doi:10.1098/rstb.2013.0264.
  3. Beaugrand, G.; Mackas, D.; Goberville, E. Applying the concept of the ecological niche and a macroecological approach to understand how climate influences zooplankton: advantages, assumptions, limitations and requirements. Progress in Oceanography 2013, 111, 75-90, doi:10.1016/j.pocean.2012.11.002.
  4. Beaugrand, G.; Goberville, E.; Luczak, C.; Kirby, R.R. Marine biological shifts and climate. Proceedings of the Royal Society B: Biological Sciences 2014, 281, 20133350, doi:10.1098/rspb.2013.3350.
  5. Beaugrand, G.; Kirby, R.R. Quasi-deterministic responses of marine species to climate change. Climate Research 2016, 69, 117-128, doi:10.3354/cr01398.
  6. Beaugrand, G.; Balembois, A.; Kléparski, L.; Kirby, R.R. Addressing the dichotomy of fishing and climate in fishery management with the FishClim model. Communications Biology 2022, 5, 1146, doi:10.1038/s42003-022-04100-6.
  7. Beaugrand, G.; Kirby, R.R. How do marine pelagic species respond to climate change? Theories and observations. Annual Review of Marine Science 2018, 10, 169–197.
  8. Beaugrand, G.; Edwards, M.; Raybaud, V.; Goberville, E.; Kirby, R.R. Future vulnerability of marine biodiversity compared with contemporary and past changes. Nature Climate Change 2015, 5, 695-701, doi:10.1038/NCLIMATE2650.
  9. Beaugrand, G.; Conversi, A.; Atkinson, A.; Cloern, J.; Chiba, S.; Fonda-Unami, S.; Kirby, R.R.; Greene, C.G.; Goberville, E.; Otto, S.A.; et al. Prediction of unprecedented biological shifts in the global ocean. Nature Climate Change 2019, 9, 237-243.
  10. Beaugrand, G.; Kirby, R.R.; Goberville, E. The mathematical influence on global patterns of biodiversity. Ecology and Evolution 2020, 10, 6494-6511, doi: 10.1002/ece3.6385.
  11. Beaugrand, G.; Luczak, C.; Goberville, E.; Kirby, R.R. Marine biodiversity and the chessboard of life Plos One 2018, 13, e0194006, doi:https://doi.org/10.1371/journal.pone.0194006.
  12. Beaugrand, G.; Kirby, R.R. How do marine species respond to climate change? Theories and observations. Annual Review of Marine Sciences 2018, 10, 169–197, doi:10.1146/121916-063304.
  13. Hellweger, F.L.; Van Sebille, E.; Fredrick, N.D. Biogeographic patterns in ocean microbes emerge in a neutral agent-based model. Science 2014, 345, 1346-1349.
  14. Lladó Fernández, S.; Větrovský, T.; Baldrian, P. The concept of operational taxonomic units revisited: genomes of bacteria that are regarded as closely related are often highly dissimilar. Folia Microbiologica 2018, 64, 19-23.
  15. Cohan, F.M.; Koeppel, A.F. The origins of ecological diversity in prokaryotes. Current Biology 2008, 18, R1024-R1034, doi:10.1016/j.cub.2008.09.014.
  16. Suen, G.; Goldman, B.S.; Welch, R.D. Predicting prokaryotic eological niches using genome sequence analysis. PLoS ONE 2007, 2, e743, doi:10.1371/journal.pone.0000743.
  17. Hughes Martiny, J.B.; Bohannan, B.J.M.; Brown, J.H.; Colwell, R.K.; Fuhrman, J.A.; Green, J.L.; Horner-Devine, M.C.; Kane, M.; Krumins, J.A.; Kuske, C.R.; et al. Microbial biogeography: putting microorganisms on the map. Nature Reviews Microbiology 2006, 4, 102–112.
  18. Beaugrand, G.; Rombouts, I.; Kirby, R.R. Towards an understanding of the pattern of biodiversity in the oceans. Global Ecology and Biogeography 2013, 22, 440–449.
  19. Kléparski, L.; Beaugrand, G. The species chromatogram, a new graphical method to represent, characterise and compare the ecological niches of different species. Ecology and Evolution 2022, 12, e8830, doi:10.1002/ece3.8830.
  20. Speck, L. Caractérisation de la biodiversité marine à l’aide de la théorie METAL; Université du Littoral Côte d'Opale: Wimereux, 2022; p. 22.
  21. Goberville, E.; Hautekèete, N.-C.; Kirby, R.R.; Piquot, Y.; Luczak, C.; Beaugrand, G. Climate change and the ash dieback crisis. Scientific Report 2016, 6, 35303, doi:10.1038/srep35303.
  22. Rombouts, I.; Beaugrand, G.; Dauvin, J.-C. Potential changes in benthic macrofaunal distributions from the English Channel simulated under climate change scenarios. Estuarine, coastal and Shelf Science 2012, 99, 153-161.
  23. Raybaud, V.; Beaugrand, G.; Goberville, E.; Delebecq, G.; Destombe, C.; Valero, M.; Davoult, D.; Morin, P.; Gevaert, F. Decline in Kelp in West Europe and Climate. PLOS One 2013, 8, e66044, doi:10.1371/journal.pone.0066044.
  24. Peters, R.H. The ecological implications of body size; Cambridge University Press: Cambridge, 1983; p. 329.
  25. Damuth, J. Population density and body size in mammals Nature 1981, 290, 699-700.
  26. Damuth, J. Of size and abundance. Nature 1991, 351, 268-269.
  27. Lomolino, M.V.; Riddle, B.R.; Brown, J.H. Biogeography, 3 ed.; Sinauer Associates, Inc.: Sunderland, 2006; p. 845.
  28. Brown, J.H. On the relationship between abundance and distribution of species. The American Naturalist 1984, 124, 255-279.
  29. Norris, R.D.; Hull, P.M. The temporal dimension of marine speciation. Evolutionary Ecology 2012, 26, 393-415, doi:10.1007/s10682-011-9488-4.
  30. Briggs, J.C. Modes of speciation: marine indo-west Pacific. Bulletin of Marine Science 1999, 65, 645-656.
  31. Pierrot-Bults, A.C.; Van der Spoel, S. Speciation in macrozooplankton. In Zoogeography and diversity of plankton, Van der Spoel, S., Pierrot-Bults, A.C., Eds.; John Wiley: New York, 1979; pp. 144-167.
  32. Paolo Momigliano; Henri Jokinen; Antoine Fraimout; Florin, A.-B.; Norkko, A.; Merilä, A. Extraordinarily rapid speciation in a marine fish. Proceedings of the National Academy of Sciences of the United States of America 2017, 114, 6074–6079.
  33. De Visser, J. Transition zones and salp speciation. In Pelagic biogeography, Pierrot-Bults, A.C., Van der Spoel, S., Zahuranec, B.J., Johnson, R.K., Eds.; Unesco: Paris, 1985; pp. 266-269.
  34. Knowlton, N. Sibling species in the sea. Annual Review of Ecology, Evolution, and Systematics 1993, 24, 189-216.
  35. Jablonski, D.; Roy, K.; Valentine, J.W. Out of the Tropics: evolutionary dynamics of the latitudinal diversity gradient. Science 2006, 314, 102-106.
  36. Dowle, E.J.; Morgan-Richards, M.; Trewick, S.A. Molecular evolution and the latitudinal biodiversity gradient. Heredity 2013, 110, 501–510, doi:10.1038/hdy.2013.4.
  37. Benton, M.J. Diversification and extinction in the history of life. Science 1995, 268, 52-58.
  38. Morlon, H. Phylogenetic approaches for studying diversification. Ecology Letters 2014, 17, 508–525.
  39. Berta, A.; Adam, P. The evolutionary biology of pinnipeds. In Secondary adaptation of tetrapods to life in the water de Buffrenil, V., Mazin, J.-M., Eds.; Verlag Dr Frederich Pfeil: Munchen Germany, 2001; pp. 235-260.
  40. Wiens, J.J.; Donoghue, M.J. Historical biogeography, ecology and species richness. Trends in Ecology and Evolution 2004, 19, 639-644.
  41. Crisp, M.D.; Cook, L.G. Phylogenetic niche conservatism: what are the underlying evolutionary and ecological causes? New Phytologist 2012, 196, 681-694.
  42. Hawkins, B.A.; Porter, E.E. Does herbivore diversity depend on plant diversity? The case of California butterflies. The American Naturalist 2003, 161, 40-49.
  43. MacArthur, R.H.; Wilson, E.O. The theory of island biogeography; Princeton University Press: Princeton 1967.
  44. Hubbell, S.P. The unified neutral theoy of biodiversity and biogeography; Princeton University Press: Princeton, 2001.
  45. Brown, J.H.; Gillooly, J.F.; Allen, A.P.; Savage, V.M.; West, G.B. Toward a metabolic theory of ecology. Ecology 2004, 85, 1771-1789.
  46. Gillooly, J.; Allen, A.P.; West, G.B.; Brown, J.H. The rate of DNA evolution: effects of body size and temperature on the molecular clock. Proceedings of the National Academy of Sciences of the United States of America 2005, 102, 140-145.
  47. Dobzhansky, T. Nothing in biology makes sense except in the light of evolution. The American Biology Teacher 1973, 35, 125-129.

Reviewer 4 Report

Summary

This review article summarizes recent work using the METAL theory and how it can be used to describe several biodiversity patterns. The strength of this review is pulling together results from previous work that use the same framework to emphasize the broad applicability of the METAL theory. This is useful for readers who may not be familiar with all of these papers, or who perhaps have seen one or two papers referring to METAL.

General comments

I think there are a few ways that this review could be made more valuable to readers who are less familiar with other papers using the METAL theory. In general, I think the review could define METAL more clearly and earlier in the article, emphasize the difference to other niche based species distribution modelling approaches, and add more detail to the individual sections summarizing previous work so that they can stand on their own.

The first three sections of this review nicely motivate the need for a more unified theory to explain patterns in ecology. However, given that the focus of this review is METAL, I feel that a brief introduction to the theory in earlier than section 4 would be helpful to orient the reader. Additionally, and perhaps more importantly, I think the explanation of METAL from Ln 136-152 could be made clearer. It isn’t clear to me from this paragraph what is new about METAL rather than explaining the ecological niche. After looking through some explanations of METAL in the references this was clearer to me, but I feel that it could be more clear here in a self-contained way.

Another aspect I think could be improved in the introductory sections is referring to other species distribution models that use the concept of the niche. There is certainly other work that uses the idea of the Hutchinson niche for determining species ranges in the environment, and I think this review would benefit from stating explicitly how METAL is different. I think this would also improve the article as a review article as it would connect METAL to other literature more clearly.

Overall, I felt that I followed each part of Section 4, but it wasn’t always clear what was part of the METAL theory and what was explaining other existing frameworks. I think this could be improved given the importance of this section to the review.

Sections 5-7 review existing literature and I find these sections well organized and clear in describing an observed pattern and stating that the framework presented here can explain these patterns. The main thing in these sections that I think would benefit readers is adding additional detail about the specific models, and in general more detailed explanations about how METAL is used to explain the patterns. For example, I did not find that the explanation from Ln 289-297 was enough for me to understand Figure 7 and had to go to the reference in the figure caption to understand how this figure was generated. I think additional explanation throughout these sections would help this review stay more self-contained and make it easier for readers to see how the METAL theory is applied.

A more specific point in section 5 is related to the explanation of Figure 5. When these niches are created, are they always fixed to have abundance 1? Another way to create these niches would be to fix the area of the niche, rather than to fix the height. It’s possible I missed something here, but if the assumption is that all niches have fixed amplitude I think that should be stated more explicitly, as it means that pseudospecies with narrow niches can never have equal abundance to those with broad niches.

Section 6 I also felt lacked enough detail to be self-contained in this review. After reading this section, it isn’t clear to me what the mathematical constraint on the number of species is, or where S comes from in Figure 8. I think given the argument that this is a mathematical constraint, some math in this section would help with understanding and precision. The sentence in Ln 331 beginning with “Since the upper limits…” I think could be edited for clarity, as it seems to be key to explaining the patterns of biodiversity. I’m also not sure why the rate of diversification is a parameter at all in this model, given the explanation of the model in the previous section doesn’t seem to include this as a parameter.

In section 7, I think the final line (388-389) could use additional support to explicitly explain how METAL theory suggests this, rather than just a reference.

Finally, I think this article would benefit from a little discussion about the future of METAL theory. Where else can this theory be applied? How can it be extended, and how else can it be tested? For example, what would happen if more variables than temperature and precipitation are included? Given the discussion in Section 7 (Ln 408-410), this would make me think that there would be more available niches and thus more species. Given that there are more than two constraining variables for real species, would METAL predict even more species as more constraints are added? Or is there some way that it accounts from the importance of different environmental constraints? Also, what would happen if the niches were not Gaussian? For temperature for example, we know thermal performance curves are not Gaussian. Or, as mentioned in one of the references, what if species interactions were included? I’m sure there are many other ways this theory could be extended and I don’t think it’s important to address any/all of these specific examples, but I do think the review would benefit from a bit of speculation and discussion near the end.

Thank you for the interesting read! I enjoyed the opportunity to look through some of these references and the organization of this review really facilitated that. I think the idea of a more unified framework to address these large-scale patterns is important, and I think this review does a nice job tying together the METAL literature.

Specific comments

-       I think another title would help prospective readers find this review more easily. To me, the article focusses on METAL and so could include that in the title. Additionally, given that the body of the text does not contain any mathematics, I think the “Mathematical constraints” part of the title is possibly a little misleading.

-       The abstract and simple summary both present METAL as a “new theory”, however the references on Ln 142 where METAL is introduced go back a decade. I think the summary and abstract should refer to METAL as a recent theory, rather than a new theory, and frame the article as a review of the theory and its applications. This is especially true for the wording on Ln 10 and Ln 20-21. I think the abstract could be a bit more clear on what is new in this article versus what is being reviewed that has been previously published. This will make it more clear this is a review article for readers browsing through abstracts.

-       This is perhaps a little too detailed a point, but I’m not sure about the statement in Ln 68-70 that constellations were the first patterns of variability. In the context of this review, I think the patterns being discussed should have some natural underlying law, whereas constellations are perceived patterns with no underlying meaning. Additionally, constellations are not the same across cultures, and so choosing Ptolemy as an example seems odd to me.

-       I’m not sure what is meant by “In our latitudes” on Ln 93. Is this the latitude of the author? Perhaps a rough latitude marker here would make this clearer.

-       Figure 1: I’m not sure if there is significance to the two drawings of copepods with different facing at the top of this figure. Are the two columns significant in some way? I think these subfigures could be reorganized to correspond more closely with the point being made in the caption where they are grouped according to where they are present.

-       As a first time reader it wasn’t immediately clear to me how Ln 205 to Ln 217 tie together with the rest of this section. I think the idea is that taking an inventory of the number of species and how they interact is important to answer the questions outlined previously, but I think this could be a bit more concise if that is the case. Given that this paragraph starts by discussing how METAL can be used to explain this patterns, I think the focus should stay on METAL, or there should be a paragraph break. Perhaps these lines could go elsewhere in the introduction?

-       Figure 7 cites Reference 24 at the end to say that the figure is adapted from that reference, and I think this reference is key to Section 5. I think there should be a reference to this paper somewhere earlier in this section to make it more clear that this is an important reference.

-       Similarly, Figure 8 cites Reference 25 at the end, and again I think putting this reference earlier in this section would be helpful for readers.

-       A small note for Ln 398/399: I think this should say “Why are there more terrestrial species than marine species?” and “Why is the number of terrestrial species higher than the number of marine species?”, or similar. I don’t think the number of terrestrial species is more important, just larger.

-       Note that there are two “Section 7”s, which I think is just a small error.

-       A small note, but there is some French in a few of the figures that I think needs to be translated.

Author Response

Reviewer 4

Reviewer 4 said “This review article summarizes recent work using the METAL theory and how it can be used to describe several biodiversity patterns. The strength of this review is pulling together results from previous work that use the same framework to emphasize the broad applicability of the METAL theory. This is useful for readers who may not be familiar with all of these papers, or who perhaps have seen one or two papers referring to METAL.”

Thank you.

Reviewer 4 said “I think there are a few ways that this review could be made more valuable to readers who are less familiar with other papers using the METAL theory. In general, I think the review could define METAL more clearly and earlier in the article, emphasize the difference to other niche based species distribution modelling approaches, and add more detail to the individual sections summarizing previous work so that they can stand on their own.”

I agree. I now mention species distribution models briefly in the revision page 7 lines 231-237:

“Also known as species distribution models (SDMs) or bioclimatic envelope models [1,2], METAL integrates ecological niche models (ENMs) in its framework. ENMs primarily assessed the realised niche (not the fundamental niche) based on past or contemporary species spatial distribution and the knowledge of some environmental variables. They then used the realised niche to project the likely distribution of a species in the past, present or future. ENMs have been extensively applied to project future species spatial distributions in the context of global climate change [3-6].”

Reviewer 4 said “The first three sections of this review nicely motivate the need for a more unified theory to explain patterns in ecology. However, given that the focus of this review is METAL, I feel that a brief introduction to the theory in earlier than section 4 would be helpful to orient the reader. Additionally, and perhaps more importantly, I think the explanation of METAL from Ln 136-152 could be made clearer. It isn’t clear to me from this paragraph what is new about METAL rather than explaining the ecological niche. After looking through some explanations of METAL in the references this was clearer to me, but I feel that it could be more clear here in a self-contained way.”

I agree. I have added a paragraph at the end of Section 1 page 2 lines 69-90:

“In this review, I present the MacroEcological Theory on the Arrangement of Life (METAL), a theory that proposes that biodiversity is strongly influenced by the climatic and the environmental regime in a deterministic manner (https://biodiversite.macroecologie.climat.cnrs.fr). This influence mainly occurs through the interactions between the ecological niche of species sensu Hutchinson (i.e. the range of a species tolerance when several factors are taken simultaneously) and the climate and  the environment [7]. The niche-environment interaction is therefore a fundamental interaction in ecology that enables one to predict and unify (i) at a species level local changes in abundance, species phenology, and biogeographic range shifts, and (ii) at a community level, the arrangement of biodiversity in space and time as well as long-term community/ecosystem shifts, including regime shifts [8-17]. This theory offers a way to make testable ecological and biogeographical predictions to understand how life is organized, and how it responds to global environmental changes [18]. More specifically here, I show how METAL helps understanding (i) why there are more species at low latitudes than at the poles, (ii) why the peak of biodiversity is located at mid-latitudes in the oceanic domain and at the equator in the terrestrial domain and (iii) finally why there are more terrestrial than marine species despite the fact that biodiversity has emerged in the oceans. METAL has not been tested on prokaryotes (Bacteria and Archaea) yet because the species concept is fuzzy in this group, being replaced by Operational Taxonomic Unit (i.e. taxa defined by molecular data analysis) [19,20]. Moreover, the ecological niche of prokaryotes can be more diverse and extreme, especially for Archaea [21,22] and their geographical ranges can be wide [23]. Therefore, all ecological principles examined in this review are only relevant for eukaryotes.”

Reviewer 4 said “Another aspect I think could be improved in the introductory sections is referring to other species distribution models that use the concept of the niche. There is certainly other work that uses the idea of the Hutchinson niche for determining species ranges in the environment, and I think this review would benefit from stating explicitly how METAL is different. I think this would also improve the article as a review article as it would connect METAL to other literature more clearly.”

I think this is indeed important. I have added a few sentences in the revision page line :

“The unification of these phenomena is obtained by using the concept of the ecological niche of Hutchinson [24,25], which constitutes the elementary macroscopic brick of the theory, giving meaning and coherence to all phenomena, patterns of variability or events cited above (Figure 3). METAL considers the fundamental niche (i.e. the niche without the influence of species interaction) and current models do not explicitly include the influence of biotic interaction yet [12,16]. The niche can be divided into five components: (i) climatic, (ii) physico-chemical, (iii) substrate, or trophic with (iv) dietary and (v) resource concentration components [10,12,26]. It integrates all environmental conditions where a species’ individual can ensure its homeostasis, grow and reproduce. A species’ niche therefore includes phenotypic plasticity, encompassing polyphenism and reaction norm (i.e. a species niche integrates the niches of all individuals of that species).

Therefore, the niche-environment interaction is considered to be a fundamental interaction in biology that explains and unify a large number of patterns observed in ecology, biogeography and climate change biology [18]. This occurs because the genome controls many processes at infraspecific organizational levels (e.g. molecular processes) that affect physiological and morphological traits that in turn influence individual performance and fitness and finally determine the ecological niche of a species [18,27]. The use of the niche makes it possible (i) to implicitly consider these infraspecific processes without having to model them and (ii) to integrate the emergence of new biological properties impossible to anticipate from the property of the individual parts when crossing one or several organizational levels (here from the molecular to the specific level)(Text Box 1) [28,29].

Also known as species distribution models (SDMs) or bioclimatic envelope models [1,2], METAL integrates ecological niche models (ENMs) in its framework. ENMs primarily assessed the realised niche (not the fundamental niche) based on past or contemporary species spatial distribution and the knowledge of some environmental variables. They then used the realised niche to project the likely distribution of a species in the past, present or future. ENMs have been extensively applied to project future species spatial distributions in the context of global climate change [3-6].”

Therefore, the niche-environment interaction is considered to be a fundamental interaction in biology that explains and unify a large number of patterns observed in ecology, biogeography and climate change biology [18]. This occurs because the genome controls many processes at infraspecific organizational levels (e.g. molecular processes) that affect physiological and morphological traits that in turn influence individual performance and fitness and finally determine the ecological niche of a species [18,27]. The use of the niche makes it possible (i) to implicitly consider these infraspecific processes without having to model them and (ii) to integrate the emergence of new biological properties impossible to anticipate from the property of the individual parts when crossing one or several organizational levels (here from the molecular to the specific level)(Text Box 1) [28,29].”

Reviewer 4 said “Overall, I felt that I followed each part of Section 4, but it wasn’t always clear what was part of the METAL theory and what was explaining other existing frameworks. I think this could be improved given the importance of this section to the review.”

Fully agree. This part has been improved page 9 line 281 in new section 5.1. entitled “A brief overview of the main hypotheses or theories that have attempted to explain biodiversity patterns”.

Reviewer 4 said “Sections 5-7 review existing literature and I find these sections well organized and clear in describing an observed pattern and stating that the framework presented here can explain these patterns. The main thing in these sections that I think would benefit readers is adding additional detail about the specific models, and in general more detailed explanations about how METAL is used to explain the patterns. For example, I did not find that the explanation from Ln 289-297 was enough for me to understand Figure 7 and had to go to the reference in the figure caption to understand how this figure was generated. I think additional explanation throughout these sections would help this review stay more self-contained and make it easier for readers to see how the METAL theory is applied.”

I fully agree. I have improved this part and I have added a new paragraph page 13 lines 402-434:

“Examples from Figure 6 show that a niche can lead to more pseudospecies in the Northern than in the Southern Hemisphere (Figures 6b-d); this is due to the current location of continents that acts as a barrier against gene flux, triggering more allopatric speciation in the Northern than the Southern Hemisphere (towards high latitudes). When the thermal amplitude is larger, a single niche leads to less pseudospecies; e.g. only one in Figure 6a in each hemisphere. Moreover, the current configuration (i.e. south to north configuration) of the continents also enables more pseudospecies to emerge in the tropics (Figures 6g-h), especially when the pseudospecies are stenoecious (Figure 6e versus Figure 6f). Note that parapatric and sympatric speciations are not accounted for in this example. Allopatric speciation is thought to be a widespread mode of speciation in the marine environment, despite more and more evidence that other modes of speciation might also be play a role [30]. Parapatric speciation is thought to be possible in the ocean [31-33]. Clinal parapatric speciation has been suggested for salps and some benthic species [30,34]. Sympatric speciation might also be frequent for marine invertebrates [35].

“To reproduce the large-scale arrangement of biodiversity, we can build a model that first creates millions of niches, which then allow pseudospecies to establish themselves in a given region so long as environmental fluctuations are suitable [7,16,17,36]; this approach is especially useful since we have only inventoried 9% of marine and 14% of terrestrial biodiversity and we know little about the biology of most species (see Section 5.1) A niche can give rise to several species if individuals from different regions never come into contact (e.g. Figure 6f) [16]. Pseudospecies are gradually colonizing the terrestrial and marine environment (surface and bottom). During the simulations, the species gradually organize themselves into communities and the biodiversity, more precisely here the number of species in a given region, is reproduced (Figure 7). These numerical experiments (or simulations) correctly reconstruct large-scale biodiversity patterns as they are observed nowadays for a large number of taxonomic groups in the terrestrial and marine environment (e.g. crustaceans, fish, cetaceans, plants, birds) [16]. The biodiversity maps for the ocean floors (Figures 7c and 7f) remain provisional as few observations have been made to date to confirm these predictions [16]. The model also reproduces well past biodiversity patterns of the Last Glacial Maximum and the mid-Pliocene (e.g. foraminifera) as well as for the Ordovician (e.g. Acritarchs) [14,37].”

Please also note that I have made substantial changes in other parts of the ms to consider this comment.

Reviewer 4 said “A more specific point in section 5 is related to the explanation of Figure 5. When these niches are created, are they always fixed to have abundance 1? Another way to create these niches would be to fix the area of the niche, rather than to fix the height. It’s possible I missed something here, but if the assumption is that all niches have fixed amplitude I think that should be stated more explicitly, as it means that pseudospecies with narrow niches can never have equal abundance to those with broad niches.”

This is an excellent point. I have clarified this page 12 lines 393-408:

“We can thus create a multitude of pseudospecies by varying the optimum and the ecological amplitude of each niche dimension. Figure 6 shows the creation of marine pseudospecies from a simple Gaussian thermal niche  [8,10]. Note that different types of niches can be used from rectangular to trapezoidal [12,36] and from logistic to Beta distribution [12,14], symmetrical or asymmetrical [12], parametric or nonparametric [10]. Moreover, the niche can be fully multidimensional [38], including nutrients, solar radiation or mixed layer depth for phytoplankton, bathymetry and sediment type for fish, soil pH and composition for plants [4,5,12,38-40]. So far, most METAL simulations were based on niches that vary between 0 (i.e. absence of a species for a given environmental regime) to 1 (i.e. highest abundance, or presence in case of a rectangular niche). Therefore, all species can reach the same level of maximum abundance. Although this assumption may possibly hold for a clade composed of species with a similar size [41-43], this is not so for a group that exhibits large size variability (e.g. mammals). Note, however, that this assumption does not affect biodiversity when the selected indicator is species richness (see below).

Reviewer 4 said “Section 6 I also felt lacked enough detail to be self-contained in this review. After reading this section, it isn’t clear to me what the mathematical constraint on the number of species is, or where S comes from in Figure 8. I think given the argument that this is a mathematical constraint, some math in this section would help with understanding and precision.

I have better explain the concept page 15 lines 475-495:

“The reconstruction of large-scale biodiversity patterns observed in nature is possible because the niche-climate interaction generates a mathematical constraint on the number of species that can establish in a given region [17]. We have named this constraint the great chessboard of life(Figure 8)[17].This particular chessboard has a number of geographic squares that correspond to different regions (marine or terrestrial). Each square of the chessboard is composed of sub-squares, which represent the number of climatic niches that determines the maximum number of species that can colonise a square (i.e. a region). Only one species can establish in a sub-square (i.e. a climatic niche) of the chessboard according to the competitive exclusion principle of Gause [44].”

See also the two modified paragraphs in my answer below.

Reviewer 4 said “The sentence in Ln 331 beginning with “Since the upper limits…” I think could be edited for clarity, as it seems to be key to explaining the patterns of biodiversity. I’m also not sure why the rate of diversification is a parameter at all in this model, given the explanation of the model in the previous section doesn’t seem to include this as a parameter.”

I have clarified the text page 15 lines 487-503. I say:

“The number of potential niches fixes an upper limit on the number of species that can colonise a given region by speciation or immigration [17]. Few species can colonise areas located towards the minimum and maximum limits of temperature and precipitation [16]. The choice of these minimum and maximum values in the METAL models is therefore important because it affects the results [36]. In marine polar areas, corresponding nowadays to the lowest limit of temperature, the number of species that can establish are fundamentally limited since two species having the same niche cannot coexist at the same time and in the same place[44].

At low latitudes, since the theoretical upper limits are not observed nowadays (i.e. the upper limit for temperature is frequently fixed to 44°C, for a justification of the threshold see ref. [36]), terrestrial biodiversity is maximum at the equator and marine biodiversity in subtropical regions (Figure 7) [14,16,36,37]. (I will come back to this point in Section 7.).”

I have also clarified this point in the revision page 16 lines 523-526:

“The biogeographical constraints (i.e. limited and elevated number of niches in high and low latitudes, respectively) imposed by the chessboard on biodiversity may be quickly detectable because clade diversification (not implemented in this METAL model) takes place relatively rapidly at a geological time scale [45].”

Reviewer 4 said “In section 7, I think the final line (388-389) could use additional support to explicitly explain how METAL theory suggests this, rather than just a reference.”

I agree this was not clear. I have improved the sentence page 18 lines 636-640:

“In the context of current climate change, the use of the METAL theory also suggests that the contrast between regions of low and high biodiversity may diminish towards the end of the century because a rise in temperature over permanently stratified regions (e.g. tropics and subtropics) reduces surface biodiversity whereas an augmentation in temperature over temperate and polar regions increases biodiversity [14].”

Reviewer 4 said “Finally, I think this article would benefit from a little discussion about the future of METAL theory. Where else can this theory be applied? How can it be extended, and how else can it be tested? For example, what would happen if more variables than temperature and precipitation are included? Given the discussion in Section 7 (Ln 408-410), this would make me think that there would be more available niches and thus more species. Given that there are more than two constraining variables for real species, would METAL predict even more species as more constraints are added? Or is there some way that it accounts from the importance of different environmental constraints? Also, what would happen if the niches were not Gaussian? For temperature for example, we know thermal performance curves are not Gaussian. Or, as mentioned in one of the references, what if species interactions were included? I’m sure there are many other ways this theory could be extended and I don’t think it’s important to address any/all of these specific examples, but I do think the review would benefit from a bit of speculation and discussion near the end.”

I agree. I am writing a book on the METAL theory and many of the ideas provided by the reviewer have been explored. I do not think this is the right place to discuss about future improvements. However, I have added the following sentences to consider reviewer’s comments:

Page 12 lines 397-401:

“Note that different types of niches can be used from rectangular to trapezoidal [12,36] and from logistic to Beta distribution [12,14], symmetrical or asymmetrical [12], parametric or nonparametric [10]. Moreover, the niche can be fully multidimensional [38], including nutrients, solar radiation or mixed layer depth for phytoplankton, bathymetry and sediment type for fish, soil pH and composition for plants [4,5,12,38-40].”

Page 21 lines 739-741:

In addition to other key ecological factors discussed in Section 5, the METAL theory should therefore consider more explicitly some key evolutionary processes in the future.

Reviewer 4 said “Thank you for the interesting read! I enjoyed the opportunity to look through some of these references and the organization of this review really facilitated that. I think the idea of a more unified framework to address these large-scale patterns is important, and I think this review does a nice job tying together the METAL literature.”

Thank you for your comments that have helped improving the paper.

Reviewer 4 said “ I think another title would help prospective readers find this review more easily. To me, the article focusses on METAL and so could include that in the title. Additionally, given that the body of the text does not contain any mathematics, I think the “Mathematical constraints” part of the title is possibly a little misleading.”

I have modified the title as follows:

Towards an understanding of large-scale biodiversity patterns on land and in the sea”

Reviewer 4 said “The abstract and simple summary both present METAL as a “new theory”, however the references on Ln 142 where METAL is introduced go back a decade. I think the summary and abstract should refer to METAL as a recent theory, rather than a new theory, and frame the article as a review of the theory and its applications. This is especially true for the wording on Ln 10 and Ln 20-21. I think the abstract could be a bit more clear on what is new in this article versus what is being reviewed that has been previously published. This will make it more clear this is a review article for readers browsing through abstracts.”

First, I have changed “new” by “recent”. I have also improved the abstract as follows:

“This review presents a recent theory named ‘MacroEcological Theory on the Arrangement of Life’ (METAL). This theory is based on the concept of the ecological niche and shows that the niche-environment (including climate) is fundamental to explain many phenomena observed in nature from the individual to the community organisational level (e.g. phenology, biogeographical shifts and community arrangement and reorganization, gradual or abrupt). The application of the theory in climate change biology and individual and species ecology has been presented elsewhere. In this review, I show how METAL explains why there are more species at low than high latitudes, why the peak of biodiversity is located at mid-latitudes in the oceanic domain and at the equator in the terrestrial domain and finally why there are more terrestrial than marine species despite the fact that biodiversity has emerged in the oceans. I postulate that the arrangement of planetary biodiversity is mathematically constrained, a constraint called the great chessboard of life, which determines the maximum number of species that may colonise a given region or domain. This theory also makes it possible to reconstruct past biodiversity and understand how biodiversity could be reorganized in the context of anthropogenic climate change.”

Reviewer 4 said “This is perhaps a little too detailed a point, but I’m not sure about the statement in Ln 68-70 that constellations were the first patterns of variability. In the context of this review, I think the patterns being discussed should have some natural underlying law, whereas constellations are perceived patterns with no underlying meaning. Additionally, constellations are not the same across cultures, and so choosing Ptolemy as an example seems odd to me.”

Agree. This part has been removed from the revision.

Reviewer 4 said “ I’m not sure what is meant by “In our latitudes” on Ln 93. Is this the latitude of the author? Perhaps a rough latitude marker here would make this clearer.”

I agree. I have clarified this point in the revision. Page 4, line 21, I have replaced ‘In our latitude’ by ‘In temperate ecosystems (e.g. the North Sea)’…

Reviewer 4 said “ Figure 1: I’m not sure if there is significance to the two drawings of copepods with different facing at the top of this figure. Are the two columns significant in some way? I think these subfigures could be reorganized to correspond more closely with the point being made in the caption where they are grouped according to where they are present.”

The two drawings being identical, I have removed one and reorganised the remaining at the centre of the figure. See the figure in the revision.

Reviewer 4 said “ As a first time reader it wasn’t immediately clear to me how Ln 205 to Ln 217 tie together with the rest of this section. I think the idea is that taking an inventory of the number of species and how they interact is important to answer the questions outlined previously, but I think this could be a bit more concise if that is the case. Given that this paragraph starts by discussing how METAL can be used to explain this patterns, I think the focus should stay on METAL, or there should be a paragraph break. Perhaps these lines could go elsewhere in the introduction?”

I have improved the paragraph page 9 lines 258-279. I say:

“The use of the METAL theory in the context of climate change biology has been presented elsewhere [18]. In this review, I show how the niche-environment interaction generates a mathematical constraint on the large-scale arrangement of biodiversity and explains why there are more species on land than in the marine realm. To make progress on these questions, the scientific community continues to collect and inventory species and to study their biology [46]. A recent study suggested that the number of terrestrial and marine species could be 8,740,000 and 2,210,000, respectively [47]. Because the scientific team estimated that 1,233,500 species had been inventoried in the terrestrial environment and 193,756 in the marine environment (bottom and surface), this suggests that between 9 (marine species) and 14% (terrestrial species) of species has been named and described so far. Ecologists investigate the multiple interactions of these species with the environment, including the climate, but also biotic interactions, an essential prerequisite for understanding their spatial distributions (biogeography), their temporal patterns of reproduction (phenology) and their fluctuations from seasonal to centenary and millennial , as well as changes occurring at a geological time scale (ecology, paleoecology and bioclimatology)[37,48-54].”

Reviewer 4 said “ Figure 7 cites Reference 24 at the end to say that the figure is adapted from that reference, and I think this reference is key to Section 5. I think there should be a reference to this paper somewhere earlier in this section to make it more clear that this is an important reference.”

Yes, I think this is an excellent suggestion. I have improved the paragraph page 13 lines 416-437:

“To reproduce the large-scale arrangement of biodiversity, we can build a model that first creates millions of niches, which then allow pseudospecies to establish themselves in a given region so long as environmental fluctuations are suitable [7,16,17,36]; this approach is especially useful since we have only inventoried 9% of marine and 14% of terrestrial biodiversity and we know little about the biology of most species (see Section 5.1) A niche can give rise to several species if individuals from different regions never come into contact (e.g. Figure 6f) [16]. Pseudospecies are gradually colonizing the terrestrial and marine environment (surface and bottom). During the simulations, the species gradually organize themselves into communities and the biodiversity, more precisely here the number of species in a given region, is reproduced. Beaugrand and colleagues [16] used this approach to model the biodiversity of the terrestrial and marine realm, including the surface and the bottom of neritic and oceanic regions (Figure 7). These numerical experiments (or simulations) correctly reconstruct large-scale biodiversity patterns as they are observed nowadays for a large number of taxonomic groups in the terrestrial and marine environment (e.g. crustaceans, fish, cetaceans, plants, birds) [16]. The biodiversity maps for the ocean floors (Figures 7c and 7f) remain provisional as few observations have been made to date to confirm these predictions [16]. The model also reproduces well past biodiversity patterns of the Last Glacial Maximum and the mid-Pliocene (e.g. foraminifera) as well as for the Ordovician (e.g. Acritarchs) [14,37].”

Reviewer 4 said “ Similarly, Figure 8 cites Reference 25 at the end, and again I think putting this reference earlier in this section would be helpful for readers.”

Yes I concur. I have added a full paragraph to better explain the concept, page 15 line 451-461:

“The reconstruction of large-scale biodiversity patterns observed in nature is possible because the niche-climate interaction generates a mathematical constraint on the number of species that can establish in a given region [17]. We have named this constraint the great chessboard of life in a previous paper (Figure 8)[17].This particular chessboard has a number of geographic squares that correspond to different regions (marine or terrestrial). Each square of the chessboard is composed of sub-squares, which represent the number of climatic niches that determines the maximum number of species that can colonise regionally a square. Only one species can establish in a sub-square (i.e. a climatic niche) of the chessboard according to the competitive exclusion principle of Gause [44].”

Reviewer 4 said “A small note for Ln 398/399: I think this should say “Why are there more terrestrial species than marine species?” and “Why is the number of terrestrial species higher than the number of marine species?”, or similar. I don’t think the number of terrestrial species is more important, just larger.”

This has been changed page 19 line 638 of the revision.

Reviewer 4 said “ Note that there are two “Section 7”s, which I think is just a small error.”

This has been modified in the revision.

Reviewer 4 said “ A small note, but there is some French in a few of the figures that I think needs to be translated.”

Yes indeed. I have modified the figure in the revision.

References added in my answers

  1. Araújo, M.B.; Peterson, A.T. Uses and misuses of bioclimatic envelope modelling. Ecology 2012, 93, 1527-1539, doi:org/10.1890/11-1930.1.
  2. Thuiller, W.; Lafourcade, B.; Engler, R.; Araújo, M.B. BIOMOD - A platform for ensemble forecasting of species distributions. Ecography 2009, 32, 369-373, doi:10.1111/j.1600-0587.2008.05742.x.
  3. Goberville, E.; Beaugrand, G.; Hautekeete, N.-C.; Piquot, Y.; Luczak, C. Uncertainties in species distribution projections and general circulation models. Ecology and Evolution 2015, 5, 1100-1116, doi:10.1002/ece3.1411.
  4. Raybaud, V.; Beaugrand, G.; Goberville, E.; Delebecq, G.; Destombe, C.; Valero, M.; Davoult, D.; Morin, P.; Gevaert, F. Decline in Kelp in West Europe and Climate. PLOS One 2013, 8, e66044, doi:10.1371/journal.pone.0066044.
  5. Rombouts, I.; Beaugrand, G.; Dauvin, J.-C. Potential changes in benthic macrofaunal distributions from the English Channel simulated under climate change scenarios. Estuarine, coastal and Shelf Science 2012, 99, 153-161.
  6. Schickele, A.; Goberville, E.; Leroy, B.; Beaugrand, G.; Hattab, T.; Francour, P.; Raybaud, V. Redistribution of small pelagic fish in Europe and Climate Change. Fish and Fisheries 2021, 22, 212-225.
  7. Beaugrand, G. Marine biodiversity, climatic variability and global change.; Routledge: London, 2015; p. 474.
  8. Beaugrand, G. Theoretical basis for predicting climate-induced abrupt shifts in the oceans. Philosophical Tansactions of the Royal Society B: Biological Sciences 2015, 370 20130264, doi:10.1098/rstb.2013.0264.
  9. Beaugrand, G.; Mackas, D.; Goberville, E. Applying the concept of the ecological niche and a macroecological approach to understand how climate influences zooplankton: advantages, assumptions, limitations and requirements. Progress in Oceanography 2013, 111, 75-90, doi:10.1016/j.pocean.2012.11.002.
  10. Beaugrand, G.; Goberville, E.; Luczak, C.; Kirby, R.R. Marine biological shifts and climate. Proceedings of the Royal Society B: Biological Sciences 2014, 281, 20133350, doi:10.1098/rspb.2013.3350.
  11. Beaugrand, G.; Kirby, R.R. Quasi-deterministic responses of marine species to climate change. Climate Research 2016, 69, 117-128, doi:10.3354/cr01398.
  12. Beaugrand, G.; Balembois, A.; Kléparski, L.; Kirby, R.R. Addressing the dichotomy of fishing and climate in fishery management with the FishClim model. Communications Biology 2022, 5, 1146, doi:10.1038/s42003-022-04100-6.
  13. Beaugrand, G.; Kirby, R.R. How do marine pelagic species respond to climate change? Theories and observations. Annual Review of Marine Science 2018, 10, 169–197.
  14. Beaugrand, G.; Edwards, M.; Raybaud, V.; Goberville, E.; Kirby, R.R. Future vulnerability of marine biodiversity compared with contemporary and past changes. Nature Climate Change 2015, 5, 695-701, doi:10.1038/NCLIMATE2650.
  15. Beaugrand, G.; Conversi, A.; Atkinson, A.; Cloern, J.; Chiba, S.; Fonda-Unami, S.; Kirby, R.R.; Greene, C.G.; Goberville, E.; Otto, S.A.; et al. Prediction of unprecedented biological shifts in the global ocean. Nature Climate Change 2019, 9, 237-243.
  16. Beaugrand, G.; Kirby, R.R.; Goberville, E. The mathematical influence on global patterns of biodiversity. Ecology and Evolution 2020, 10, 6494-6511, doi: 10.1002/ece3.6385.
  17. Beaugrand, G.; Luczak, C.; Goberville, E.; Kirby, R.R. Marine biodiversity and the chessboard of life Plos One 2018, 13, e0194006, doi:https://doi.org/10.1371/journal.pone.0194006.
  18. Beaugrand, G.; Kirby, R.R. How do marine species respond to climate change? Theories and observations. Annual Review of Marine Sciences 2018, 10, 169–197, doi:10.1146/121916-063304.
  19. Hellweger, F.L.; Van Sebille, E.; Fredrick, N.D. Biogeographic patterns in ocean microbes emerge in a neutral agent-based model. Science 2014, 345, 1346-1349.
  20. Lladó Fernández, S.; Větrovský, T.; Baldrian, P. The concept of operational taxonomic units revisited: genomes of bacteria that are regarded as closely related are often highly dissimilar. Folia Microbiologica 2018, 64, 19-23.
  21. Cohan, F.M.; Koeppel, A.F. The origins of ecological diversity in prokaryotes. Current Biology 2008, 18, R1024-R1034, doi:10.1016/j.cub.2008.09.014.
  22. Suen, G.; Goldman, B.S.; Welch, R.D. Predicting prokaryotic eological niches using genome sequence analysis. PLoS ONE 2007, 2, e743, doi:10.1371/journal.pone.0000743.
  23. Hughes Martiny, J.B.; Bohannan, B.J.M.; Brown, J.H.; Colwell, R.K.; Fuhrman, J.A.; Green, J.L.; Horner-Devine, M.C.; Kane, M.; Krumins, J.A.; Kuske, C.R.; et al. Microbial biogeography: putting microorganisms on the map. Nature Reviews Microbiology 2006, 4, 102–112.
  24. Hutchinson, G.E. Concluding remarks. Cold Spring Harbor Symposium Quantitative Biology 1957, 22, 415-427.
  25. Hutchinson, G.E. An introduction to population ecology; Yale University Press: New Haven, 1978; p. 260.
  26. Caracciolo, M.; Beaugrand, G.; Hélaouët, P.; Gevaert, F.; Edwards, M.; Lizon, F.; Kléparski, L.; Goberville, E. Annual phytoplankton succession results from niche-environment interaction. Journal of Plankton Research 2021, 43, 85-102, doi:10.1093/plankt/fbaa060.
  27. Kléparski, L.; Beaugrand, G.; Edwards, M.; Schmitt, F.G.; Kirby, R.R.; Breton, E.; Gevaert, F.; Maniez, E. Morphological traits, niche-environment interaction and temporal changes in diatoms. Progress in Oceanography 2022, 201, 102747, doi:https://doi.org/10.1016/j.pocean.2022.102747.
  28. Frontier, S.; Pichot-Viale, D.; Leprêtre, A.; Davoult, D.; Luczak, C. Ecosystèmes. Structure, fonctionnement et évolution, 3 ed.; Dunod: Paris, 2004; p. 549.
  29. Morin, E. Introduction à la pensée complexe.; Editions du Seuil: Paris, 2005; p. 158.
  30. Norris, R.D.; Hull, P.M. The temporal dimension of marine speciation. Evolutionary Ecology 2012, 26, 393-415, doi:10.1007/s10682-011-9488-4.
  31. Briggs, J.C. Modes of speciation: marine indo-west Pacific. Bulletin of Marine Science 1999, 65, 645-656.
  32. Pierrot-Bults, A.C.; Van der Spoel, S. Speciation in macrozooplankton. In Zoogeography and diversity of plankton, Van der Spoel, S., Pierrot-Bults, A.C., Eds.; John Wiley: New York, 1979; pp. 144-167.
  33. Paolo Momigliano; Henri Jokinen; Antoine Fraimout; Florin, A.-B.; Norkko, A.; Merilä, A. Extraordinarily rapid speciation in a marine fish. Proceedings of the National Academy of Sciences of the United States of America 2017, 114, 6074–6079.
  34. De Visser, J. Transition zones and salp speciation. In Pelagic biogeography, Pierrot-Bults, A.C., Van der Spoel, S., Zahuranec, B.J., Johnson, R.K., Eds.; Unesco: Paris, 1985; pp. 266-269.
  35. Knowlton, N. Sibling species in the sea. Annual Review of Ecology, Evolution, and Systematics 1993, 24, 189-216.
  36. Beaugrand, G.; Rombouts, I.; Kirby, R.R. Towards an understanding of the pattern of biodiversity in the oceans. Global Ecology and Biogeography 2013, 22, 440–449.
  37. Zacaï, A.; Monnet, C.; Pohl, A.; Beaugrand, G.; Mullins, G.; Kröck, D.; Servais, T. Truncated bimodal latitudinal diversity gradient in early Paleozoic phytoplankton. Science Advances 2021, 7, eabd6709, doi:10.1126/sciadv.abd6709.
  38. Kléparski, L.; Beaugrand, G. The species chromatogram, a new graphical method to represent, characterise and compare the ecological niches of different species. Ecology and Evolution 2022, 12, e8830, doi:10.1002/ece3.8830.
  39. Speck, L. Caractérisation de la biodiversité marine à l’aide de la théorie METAL; Université du Littoral Côte d'Opale: Wimereux, 2022; p. 22.
  40. Goberville, E.; Hautekèete, N.-C.; Kirby, R.R.; Piquot, Y.; Luczak, C.; Beaugrand, G. Climate change and the ash dieback crisis. Scientific Report 2016, 6, 35303, doi:10.1038/srep35303.
  41. Peters, R.H. The ecological implications of body size; Cambridge University Press: Cambridge, 1983; p. 329.
  42. Damuth, J. Population density and body size in mammals Nature 1981, 290, 699-700.
  43. Damuth, J. Of size and abundance. Nature 1991, 351, 268-269.
  44. Gause, G.F. The struggle for coexistence; MD: Williams and Wilkins: Baltimore, 1934.
  45. Rabosky, D.L.; Hurlbert, A.H. Species richness at continental sclaes is dominated by ecological limits. The American Naturalist 2015 185, 572-583.
  46. Raven, P.H.; Johnson, G.B.; Mason, K.A.; Losos, J.B.; Singer, S.R. Biologie; Deboeck Supérieur: Louvain-la-Neuve, 2017; p. 1282.
  47. Mora, C.; Tittensor, D.P.; Adl, S.; Simpson, A.G.B.; Worm, B. How Many Species Are There on Earth and in the Ocean? PLoS Biology 2011, 9, e1001127, doi:10.1371/journal.pbio.100112.
  48. Thackeray, S.J.; Henrys, P.A.; Hemming, D.; Bell, J.R.; Botham, M.S.; Burthe, S.; Helaouet, P.; Johns, D.G.; Jones, I.D.; Leech, D.I.; et al. Phenological sensitivity to climate across taxa and trophic levels. Nature 2016, 535, 241-245, doi:10.1038/nature18608

http://www.nature.com/nature/journal/v535/n7611/abs/nature18608.html#supplementary-information.

  1. Parmesan, C.; Ryrholm, N.; Stefanescu, C.; Hill, J.K.; Thomas, C.D.; Descimon, H.; Huntley, B.; Kaila, L.; Kullberg, J.; Tammaru, T.; et al. Poleward shifts in geographical ranges of butterfly species associated with regional warming. Nature 1999, 399, 579-583.
  2. Parmesan, C.; Matthews, J. Biological impacts of climate change. In Principles of concervation biology, Groom, M.J., Meffe, G.K., Carroll, C.R., Eds.; Sinauer Associates, Inc: Sunderland, 2006; pp. 333-360.
  3. Beaugrand, G.; Brander, K.M.; Lindley, J.A.; Souissi, S.; Reid, P.C. Plankton effect on cod recruitment in the North Sea. Nature 2003, 426, 661-664.
  4. Beaugrand, G.; Reid, P.C.; Ibañez, F.; Lindley, J.A.; Edwards, M. Reorganisation of North Atlantic marine copepod biodiversity and climate. Science 2002, 296, 1692-1694.
  5. Post, E.; Peterson, R.O.; Stenseth, N.C.; McLaren, B.E. Ecosystem consequences of wolf behavioural response to climate. Nature 1999, 1999, 905-907.
  6. Poloczanska, E.S.; Brown, C.J.; Sydeman, W.J.; Kiessling, W.; Schoeman, D.S.; Moore, P.J.; Brander, K.; Bruno, J.F.; Buckley, L.B.; Burrows, M.T.; et al. Global imprint of climate change on marine life. Nature Climate Change 2013, 3, 919-925.

Round 2

Reviewer 2 Report

Well done! This revised version of the manuscript is ready for publication in MDPI.

Author Response

Thank you

Reviewer 4 Report

Overall I feel that most of my comments have been addressed. In particular, I feel that the manuscript is now organized in a way that is easier to follow for readers less familiar with METAL, and each section stands on its own much better. I have a few minor comments on the revised manuscript.

-       I appreciate the addition of the paragraph Ln 206-212 discussing other SDMs and bioclimatic envelope models. However, I still feel that this could be fleshed out a little more. I am not that familiar with these types of bioclimatic envelope models, but it is still not immediately clear to me what METAL is doing differently compared to these other models.

-       Again overall I feel that the sections are described in a way that is more self contained, but there are two areas (the next two points) where I feel additional details could still be useful for readers.

-       At line 408, looking at Ref 29, it does seem that the generation of these niches is somewhat complicated, but I think it is worth specifying if this range is for the optimum or amplitude, and if both are varied or not. Given that so many niches are created within only temperature or temperature and precipitation dimensions, I think it’s worth a little additional detail.

-       For the great chessboard analogy, I still do not understand Fig. 8 and feel that it warrants additional explanation. How is S determined in the figure? Are the different pieces on the chessboard meant to be different species, or include intraspecific variation? Is this meant to be a schematic, or is this truly how many niches are available in the different geographical ranges? And finally, what is the significance of the chosen scale? The discussion in this section I feel is much clearer, but I feel Fig. 8 itself needs more explanation.

-       I’m still not clear to what extent the constraints discussed are mathematical versus biological or climatic, but I think this is a minor detail. The way I understand the argument, these constraints arise from the number of available niches given the climate in an area. Thus the constraints are more climatic or biological than mathematical. But I think this is a minor language point and perhaps I misunderstand the intent here.

-       Several of the figures still have French in them, but this is a small edit.

Author Response

Reviewer 4

Reviewer 4 said ‘Overall I feel that most of my comments have been addressed. In particular, I feel that the manuscript is now organized in a way that is easier to follow for readers less familiar with METAL, and each section stands on its own much better. I have a few minor comments on the revised manuscript.’

Thank you.

Reviewer 4 said ‘I appreciate the addition of the paragraph Ln 206-212 discussing other SDMs and bioclimatic envelope models. However, I still feel that this could be fleshed out a little more. I am not that familiar with these types of bioclimatic envelope models, but it is still not immediately clear to me what METAL is doing differently compared to these other models.

I have improved the paragraph and have added a new sentence at the end of it. I say page 7 lines 210-219:

Also known as species distribution models (SDMs) or bioclimatic envelope models [1,2], METAL integrates ecological niche models (ENMs) in its framework. ENMs primarily focus on the realised niche, which is based on past or contemporary spatial distribution and some key environmental (including climatic) variables. They then use the realised niche to project the likely distribution of a species in the past, present or future. ENMs have been extensively applied to project future species spatial distributions in the context of global climate change [1,3-8]. METAL provides a robust scientific baseline for ENMs and shows that this niche approach can be extended to explain many different phenomena at different organisational levels and spatio-temporal scales [9,10].

Reviewer 4 said ‘Again overall I feel that the sections are described in a way that is more self contained, but there are two areas (the next two points) where I feel additional details could still be useful for readers. At line 408, looking at Ref 29, it does seem that the generation of these niches is somewhat complicated, but I think it is worth specifying if this range is for the optimum or amplitude, and if both are varied or not. Given that so many niches are created within only temperature or temperature and precipitation dimensions, I think it’s worth a little additional detail.”

I have improved the following sentences page 13 lines 415-422:

“It starts to create a large number of niches where both optimums and amplitudes with respect to temperature only for the marine realm and both temperature and precipitation for the terrestrial realm vary. Many niches (i.e. with all possible optimums and amplitudes), which can also overlap, are created (i) for temperature between -1.8°C and 44°C in both realms and (ii) for precipitation between 0 and 3,000 mm in the terrestrial realm only; the full procedure is described in ref. [29]. At the end of the procedure, there are a maximum of 101,397 and 94,299,210 niches in the marine and terrestrial realms, respectively.”

Reviewer 4 said ‘For the great chessboard analogy, I still do not understand Fig. 8 and feel that it warrants additional explanation. How is S determined in the figure? Are the different pieces on the chessboard meant to be different species, or include intraspecific variation? Is this meant to be a schematic, or is this truly how many niches are available in the different geographical ranges? And finally, what is the significance of the chosen scale? The discussion in this section I feel is much clearer, but I feel Fig. 8 itself needs more explanation.’

How S is determined in the figure is now explained in the main text and also shown in the Figure. The other questions have been considered and I have improved the first paragraph of the section accordingly. I say pages 14-15 lines 460-484:

‘The reconstruction of large-scale biodiversity patterns observed in nature is possible because the niche-climate interaction generates a mathematical constraint on the maximum number of species that can establish in a given region [30]. We have named this constraint the great chessboard of life (Figure 8)[30]. This particular chessboard has a number of geographic squares (i.e. wide squares in Figure 8) that correspond to different regions (marine or terrestrial); note that these geographical squares are limited on the figure (i.e. 6 x 8 = 48 squares) but should be higher to correctly represent the variety of environments, e.g. one every degree of latitude and longitude (i.e. 180 latitudes x 360 longitudes = 64800 squares). Each square on the chessboard is composed of sub-squares (i.e. the narrow squares in Figure 8), which represent the number of climatic niches that determines the maximum number of species that can colonise a square (i.e. a region or a wide square). Only one species can establish in a sub-square (i.e. a climatic niche) of the chessboard according to the competitive exclusion principle of Gause [158]; thereby the more sub-squares (L) in a given region, the higher the maximum number of species that an area may contain (Figure 8) S is the number of species that a square (i.e. an area) actually contains. Therefore, L represents a fundamental limit (what I call here a mathematical constraint) on species richness, even if the actual number can still vary according to other processes (see below). The different pieces on the chessboard (e.g. king, queen, pawn) symbolize the different biological properties of the species (e.g. their differences in terms of life history traits such as reproduction). Q represents niche saturation, with Q=(S/L) x 100. A saturation of 100% means that all niches or potential species that a square may contain are occupied. Biological (degree of clade origination) and climatic (repeated Pleistocene glaciations) causes influence the percentage of saturation of the niches in each geographical square so that there remains a degree of valence on the number of species present on the great chessboard of life [30].’

Reviewer 4 said ‘I’m still not clear to what extent the constraints discussed are mathematical versus biological or climatic, but I think this is a minor detail. The way I understand the argument, these constraints arise from the number of available niches given the climate in an area. Thus the constraints are more climatic or biological than mathematical. But I think this is a minor language point and perhaps I misunderstand the intent here.’

I think the reviewer has understood but his/her focus is on the niche-environment interaction while I focus on the mathematical consequences of this interaction in term of maximum species that can colonise a given area. That is why I say this is a mathematical constraint. I have kept this term because this is the main reason why species richness is constrained locally, although other factors (discussed in the review) may also play a role. I have clarified this point in the revision (see my previous answer). I say page 15 lines 475-477:

“Therefore, L represents a fundamental limit (what I call here a mathematical constraint) on species richness, even if the actual number can still vary according to other processes (see below).”

Reviewer 4 said ‘Several of the figures still have French in them, but this is a small edit.’

I am sorry. I have modified the words in Fig. 2 and 3.

Literature cited

  1. Araújo, M.B.; Peterson, A.T. Uses and misuses of bioclimatic envelope modelling. Ecology 2012, 93, 1527-1539, doi:org/10.1890/11-1930.1.
  2. Thuiller, W.; Lafourcade, B.; Engler, R.; Araújo, M.B. BIOMOD - A platform for ensemble forecasting of species distributions. Ecography 2009, 32, 369-373, doi:10.1111/j.1600-0587.2008.05742.x.
  3. Goberville, E.; Beaugrand, G.; Hautekeete, N.-C.; Piquot, Y.; Luczak, C. Uncertainties in species distribution projections and general circulation models. Ecology and Evolution 2015, 5, 1100-1116, doi:10.1002/ece3.1411.
  4. Raybaud, V.; Beaugrand, G.; Goberville, E.; Delebecq, G.; Destombe, C.; Valero, M.; Davoult, D.; Morin, P.; Gevaert, F. Decline in Kelp in West Europe and Climate. PLOS One 2013, 8, e66044, doi:10.1371/journal.pone.0066044.
  5. Rombouts, I.; Beaugrand, G.; Dauvin, J.-C. Potential changes in benthic macrofaunal distributions from the English Channel simulated under climate change scenarios. Estuarine, coastal and Shelf Science 2012, 99, 153-161.
  6. Schickele, A.; Goberville, E.; Leroy, B.; Beaugrand, G.; Hattab, T.; Francour, P.; Raybaud, V. Redistribution of small pelagic fish in Europe and Climate Change. Fish and Fisheries 2021, 22, 212-225.
  7. Peterson, A.T.; Cobos, M.E.; Jiménez-García, D. Major challenges for correlational ecological niche model projections to future climate conditions: Climate change, ecological niche models, and uncertainty. Annals of the New York Academy of Sciences 2018, 1429, 66-77.
  8. Ehrlén, J.; Morris, W.F. Predicting changes in the distribution and abundance of species under environmental change. Ecology Letters 2015, 18, 303-314.
  9. Helaouët, P.; Beaugrand, G. Physiology, ecological niches and species distribution. Ecosystems 2009, 12, 1235-1245.
  10. Beaugrand, G.; Mackas, D.; Goberville, E. Applying the concept of the ecological niche and a macroecological approach to understand how climate influences zooplankton: advantages, assumptions, limitations and requirements. Progress in Oceanography 2013, 111, 75-90, doi:10.1016/j.pocean.2012.11.002.